# NogoA-expressing astrocytes limit peripheral macrophage infiltration after ischemic brain injury in primates

Anthony G. Boghdadi [1], Joshua Spurrier[2,3,4,10], Leon Teo[1,10], Mingfeng Li [2,4], Mario Skarica[2,4], Benjamin Cao[1,5], William C. Kwan [1], Tobias D. Merson [1], Susan K. Nilsson[1,5], Nenad Sestan [2,4,6,7,8,9], Stephen M. Strittmatter [2,3,4] & James A. Bourne [1✉]

Astrocytes play critical roles after brain injury, but their precise function is poorly defined. Utilizing single-nuclei transcriptomics to characterize astrocytes after ischemic stroke in the visual cortex of the marmoset monkey, we observed nearly complete segregation between stroke and control astrocyte clusters. Screening for the top 30 differentially expressed genes that might limit stroke recovery, we discovered that a majority of astrocytes expressed RTN4A/ NogoA, a neurite-outgrowth inhibitory protein previously only associated with oligodendrocytes. NogoA upregulation on reactive astrocytes post-stroke was significant in both the marmoset and human brain, whereas only a marginal change was observed in mice. We determined that NogoA mediated an anti-inflammatory response which likely contributes to limiting the infiltration of peripheral macrophages into the surviving parenchyma.

[1] Australian Regenerative Medicine Institute, Monash University, 15 Innovation Walk, Clayton, VIC 3800, Australia. [2] Cellular Neuroscience, Neurodegeneration, and Repair Program, Yale University School of Medicine, New Haven, CT 06536, USA. [3] Department of Neurology, Yale University School of Medicine, New Haven, CT 06536, USA. [4] Department of Neuroscience, Yale University School of Medicine, New Haven, CT 06536, USA. [5] Biomedical Manufacturing CSIRO Manufacturing, Clayton, VIC 3169, Australia. [6] Department of Genetics, Yale University School of Medicine, New Haven, CT 06536, USA. [7] Department of Psychiatry, Yale University School of Medicine, New Haven, CT 06536, USA. [8] Department of Comparative Medicine, Yale University School of Medicine, New Haven, CT 06536, USA. [9] Yale Child Study Center, Yale University School of Medicine, New Haven, CT 06536, USA. [10]These authors contributed equally: Joshua Spurrier, Leon Teo. ✉email: james.bourne@monash.edu

While thrombolytic therapies have proven beneficial for ischemic stroke recovery, decades of effort to translate neuroprotective discoveries from rodent experiments to the clinical setting have been typified by numerous failures[1]. There are likely multiple explanations, but a focus on rodents for discovery research and an emphasis on neuronal responses are likely to be crucial factors. For this reason, we investigated astroglial responses in a nonhuman primate (NHP) model of ischemic stroke, which has previously been demonstrated to be more akin to the human condition[2].

It is well established that astrocytes are crucial players in the pathogenesis of brain injuries, such as ischemic stroke, yet experimental investigation has remained centered on the neuron[3]. In response to exogenous or endogenous stimuli arising from injury, disease, or aging, astrocytes are triggered into their reactive state (reactive astrocytes), characterized by drastic changes to their gene expression, morphology, and function. Although it is recognized that reactive astrocytes can exacerbate secondary neurodegeneration, recent studies have highlighted that they are a highly heterogeneous population of cells that can also provide crucial neuroprotection for surviving neurons in response to CNS pathologies[4]. Recent transcriptome analyses have clearly demonstrated the evolutionary novelties of specific CNS cell types in primates compared to rodents[5], highlighting the power of transcriptomic studies in NHPs. Here, we provide a catalog of NHP astrocyte transcriptional changes in a clinically translatable stroke model and identify putative therapeutic targets from the reactive astrocyte-mediated pathogenesis of ischemic stroke.

Single-nuclei RNA sequencing (snRNAseq) revealed near complete segregation of astrocytes between the injured and uninjured brain highlighting the extensive transcriptomic changes, mostly immune-associated, that occur as a consequence of ischemic stroke. Unexpectedly, we discovered that a neurite outgrowth inhibiting protein associated with myelinating oligodendrocytes (NogoA)[6,7] is expressed by a large population of reactive astrocytes. This specific response may provide a local secondary source of neurite outgrowth inhibition, potentially reinforcing oligodendrocyte effects. We present evidence that a collar of reactive astrocytes expressing NogoA likely play an anti-inflammatory role in the subacute stroke period to limit peripheral macrophage infiltration from the ischemic zone (adjacent to the ischemic core). Therefore, immediate post-stroke blockade of NogoA action may exacerbate brain inflammation in primate species, including humans. These data highlight the pronounced transcriptomic changes that occur in astrocytes and their response to ischemic damage.

## Results

**snRNAseq revealed near complete segregation of stroke and control astrocyte clusters.** Single-nuclei RNAseq (snRNAseq) on the 10X platform was used to profile NHP astrocytes following ischemic stroke. Adult marmosets ($n = 3$) received unilateral injections of endothelin-1 (ET-1) to the primary visual cortex (V1), with gray matter tissues encompassing all cortical layers of operculum collected 7 days later[2] (Fig. 1a; Supplementary Fig. 1a). Non-lesioned adult marmoset ($n = 3$) V1 was used as control. Single nuclei (29,537 total from the injured cohort and 8,610 total from the control cohort; Supplementary Fig. 1b) were isolated from snap frozen tissue. Following sequencing, individual nuclei underwent unsupervised clustering and were arranged by uniform manifold approximation and projection (UMAP) for visualization, with nuclei from each cohort being analyzed separately to ensure effective cell type-specific clustering in each (Fig. 1b). Clusters were assigned to neural cell types based on the expression patterns of known cell-type-specific markers (Fig. 1c, Supplementary Fig. 1c). Additional quality control on the injured cohort of marmosets was performed to demonstrate how the data clusters across the dataset (Supplementary Fig. 1d–f). We observed no obvious differences between the $n = 3$ injured marmosets across the number of genes expressed (Supplementary Fig. 1d), UMIs (Supplementary Fig. 1e) and cells captured (Supplementary Fig. 1f), although there was a larger number of ExN nuclei captured in one marmoset. This highlights the consistency and strength of the data across our injured cohort of adult marmosets.

There has already been considerable previous focus on neuronal changes following ischemic injury, so we sought to investigate the changes to astrocytes after stroke. Astrocyte nuclei profiles from both cohorts were isolated from the datasets (2,107 from the injured cohort and 594 from the control cohort; Supplementary Fig. 1b), merged, integrated, scaled and processed using the Seurat algorithm. Significant post-stroke changes were evident with near complete segregation between injured and control observed when arranged by UMAP for visualization (Fig. 1d). Astrocytes from both cohorts expressed appropriate pan-astrocyte markers (SLC1A2, SLC1A3, ALDH1L1, AQP4) and injured cohorts increased levels of pan-reactive markers (GFAP, VIM, SERPINA3, CXCL10, OSMR) (Fig. 1e). A total of 576 differentially expressed genes (DEGs) were identified between injured and control astrocytes, with significantly more upregulated than downregulated (529 upregulated versus 47 downregulated) (Fig. 1f, Supplementary Data 1). Of particular interest, the top DEGs predominantly possessed immunomodulatory roles, such as cytokine/ interferon signaling pathways and defense response to virus (Supplementary Data 1). However, within the top 30 DEGs, it was notably observed that RTN4A (NogoA), previously described as an oligodendrocyte marker[8] and known to inhibit neurite outgrowth[9,10], was significantly upregulated (Fig. 1f, highlighted in red) and expressed in a large majority of reactive astrocytes (Fig. 2a). Astrocytes were segregated based on RTN4A expression, and RTN4A- and RTN4A + astrocytes were found to share 252 injury-dependent DEGs, with 187 unique to RTN4A- and 178 unique to RTN4A + astrocytes (Fig. 1g, Supplementary Data 1). Characterization of these unique DEGs revealed distinct GO terms associated with positive regulation of synaptic plasticity and ion transport in RTN4A- astrocytes, compared to negative regulation of the leukocyte response and exocytosis in RTN4A + astrocytes (Fig. 1h, Supplementary Data 1). Further characterization of the top 100 DEGs in RTN4A- nuclei using Functional Module Detection (HumanBase) revealed four distinct functional categories (Fig. 1i, Supplementary Data 1). Remarkably, characterization of the top 100 DEGs in RTN4A + nuclei using the same technique revealed significant overlap between three out of the four functional categories, but with a distinct difference in one, which included regulation of the adaptive immune response, leukocyte trafficking, and blood–brain barrier integrity (Fig. 1j, Supplementary Data 1). These data indicate that RTN4A may be involved in functions other than those it is typically associated within the literature, including immunomodulatory reactive astrocyte functions after stroke in primates.

Genes spanning two functional categories within the RTN4A + population were used for in tissue validation of transcriptomic data (GAP43, CD44, KLF6) (Fig. 1h, highlighted in red). GAP43 was recently shown to mediate glial plasticity and promote neuroprotection[11]. The transcriptional activator KLF6 and major surface hyaluronan (HA) receptor CD44, involved in cell proliferation and cell adhesion/ migration, respectively, were previously reported to be upregulated following a middle cerebral artery occlusion[12,13]. Alongside expression of RTN4A, comprising

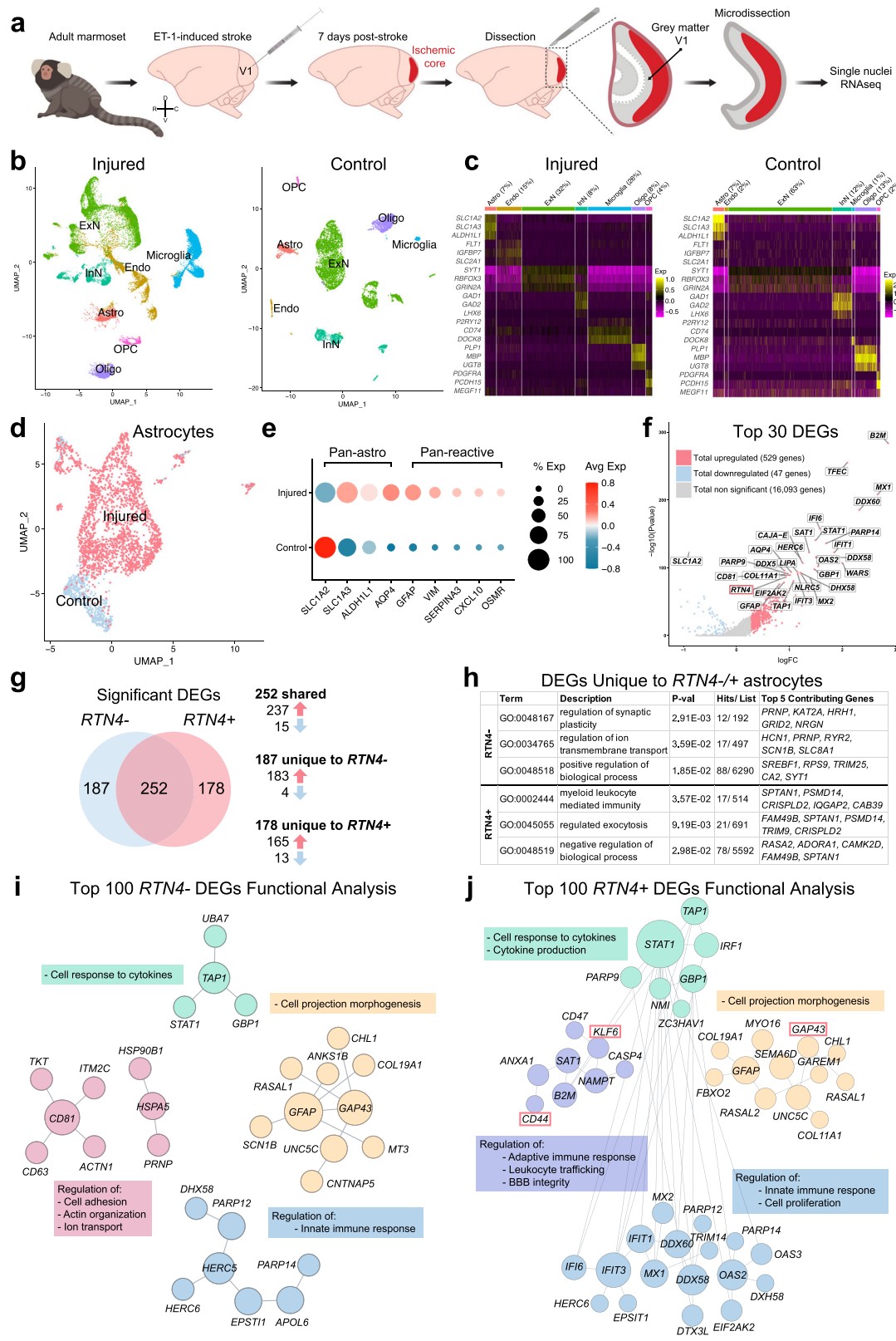

76% total astrocytes in the injured cohort and 21% in control, we observed robust expression of *GAP43, CD44* and *KLF6*, with both the percent of nuclei expressing these genes and the overall level of expression significantly increased after stroke (Fig. 2a). Subsequent immunohistochemical staining in tissues from 1-week post-ischemic stroke and control marmosets, revealed

an increased number of astrocytes colocalizing NogoA and either GAP43, KLF6, or CD44, consistent with the transcriptomic results (Fig. 2b–d). Notably, the in-tissue quantification of NogoA + / GAP43 + astrocytes (Fig. 2c) was also consistent with the numbers obtained at the nuclei level from the transcriptomic data (Fig. 2a).

**Fig. 1 Molecular classification of marmoset post-ischemic stroke reactive astrocytes. a** Schematic depicting ET-1-induced ischemic stroke in adult marmoset V1 and tissues isolated for single-nuclei 10x genomic sequencing. The injury was induced in the left hemisphere of three adult marmosets; uninjured hemispheres of three adult marmosets were used as controls for transcriptomic analyses. GM encompassing all cortical layers of V1 operculum was collected and isolated from both injured and control samples. **b** UMAP visualization of single nuclei from injured and control samples, colored by cell type. **c** Heat map colored by single nuclei gene expression of cell-type-specific markers. Heat map of additional markers used for cell-type assignment is in Supplementary Fig. 1c. **d** UMAP visualization of single astrocyte nuclei from injured and control samples, colored by cluster. **e** Average expression of pan-astrocyte and reactive astrocyte markers by condition. The size of the dots indicates the percentage of nuclei within that group that are expressing the gene at any level, while color indicates average expression from all nuclei within that group. **f** DEGs between injured and control samples; the log of the fold change between conditions is plotted on the axes. Genes were deemed significant if they exhibited >0.25 log-transformed fold change in either direction, and if the genes were expressed in >10% of nuclei in both groups. Red dots represent significantly upregulated DEGs, while blue dots represent DEGs that are significantly downregulated. Gray dots are genes with unaltered expression in between both conditions. The gene of interest, *RTN4A* (NogoA), within the top 30 DEGs is highlighted by a red bounding box. For the full DEG list, refer to Supplementary Data 1. **g** Venn diagram demonstrating shared and unique significant DEGs in *RTN4A*- and *RTN4A* + astrocytes. **h** GO analysis of unique DEGs in *RTN4A*- and *RTN4A* + astrocytes. For the full list of GO terms and genes, refer to Supplementary Data 1. **i** Characterization of the top 100 *RTN4A*- DEGs by Functional Module Detection (HumanBase), colored by functional module with summarized terms. For the full *RTN4A*- DEG list and Functional Module Detection analysis, refer to Supplementary Data 1. **j** Characterization of the top 100 *RTN4A* + DEGs by Functional Module Detection (HumanBase), colored by functional module with summarized terms. Genes of interest with concordant expression to *RTN4A*: *GAP43, KLF6,* and *CD44,* are highlight by red bounding boxes. For the full *RTN4A* + DEG list and Functional Module Detection analysis, refer to Supplementary Data 1. ET-1 endothelin-1, V1 primary visual cortex, GM gray matter, UMAP uniform manifold approximation and projection, Astro astrocyte, Endo endothelial cell, ExN excitatory neurons; InN inhibitory neurons, Oligo oligodendrocytes, OPC oligodendrocyte precursor cells, DEGs differentially expressed genes, *RTN4A* (NogoA) reticulon-4/ neurite outgrowth inhibitor A, *GAP43* growth-associated protein 43, *KLF6* kruppel-like factor 6, *CD44* cell-surface glycoprotein-44.

**Reactive astrocytes express neurite outgrowth inhibitory protein NogoA in primates post-ischemic stroke.** The best-characterized sequelae of NogoA expression after CNS injuries is inhibition of neurite outgrowth[6,7,14,15], through neuron axon–oligodendrocyte interactions. Having identified that reactive astrocytes upregulate NogoA in the marmoset 1-week post-stroke (Fig. 1f, i; 2a–c), and given that this had not been previously described in rodent literature, we sought to investigate astrocytic NogoA expression at the equivalent post-stroke time point of 3 and 7 DPI in mouse and marmoset, respectively[16–18]. Our analysis revealed fewer than 50% of astrocytes adjacent to the ischemic core in mouse cortex were NogoA + (Fig. 3a, c), with minimal and punctate labeling compared to the widespread and uniform NogoA expression observed in almost all marmoset astrocytes at 7 DPI within the ischemic zone (Fig. 3b, c). Furthermore, the fluorescence intensity of NogoA immunoreactivity in astrocytes of the mouse was negligible relative to that in the marmoset (Fig. 3d). These data are supported by previous transcriptomic analysis of reactive astrocytes where *RTN4A* remains unchanged in adult mice 3 days after transient middle cerebral artery occlusion[19,20]. Macroscopically, it is evident that the vast majority of GFAP + astrocytes in marmosets express NogoA within the ischemic zone, tapering off at the V1/ V2 border with decreasing levels of NogoA expression (Supplementary Fig. 2). Together, these data indicate that there are marked differences between mouse and marmoset in terms of NogoA expression on reactive astrocytes and that its upregulation post-stroke may be unique to primates.

To investigate the marmoset-specific NogoA+ astrocytes further, we extended our analysis to several key pathophysiological time points post-stroke. We were able to demonstrate a transient upregulation of NogoA on GFAP + reactive astrocytes for at least 2 weeks post-ischemic stroke in the marmoset (Supplementary Fig. 3). We also demonstrated that the population of NogoA+ reactive astrocytes upregulated select markers, such as GAP43 (Fig. 2b–c), CD44, and KLF6 (Fig. 2d–e), in tissue, consistent with our transcriptomic data.

To determine if the NogoA+ reactive astrocytes observed in marmosets were found following human stroke, post-mortem human cortical tissue (1-week post-ischemic stroke) was sourced from the Newcastle Brain Tissue Resource (UK). Immunolabeling revealed a subpopulation of GFAP + reactive astrocytes co-expressing NogoA adjacent to the ischemic core (Fig. 3e), consistent with our data in the marmoset at the same pathophysiological time point. Importantly, the expression profile of NogoA in human reactive astrocytes is consistent with the marmoset and unlikely to be accumulation of myelin debris[21]. These results confirm that the expression of NogoA was associated with astrocytes at an identical time point post-ischemic stroke in both marmoset and human.

Following CNS injury, myelin breakdown occurs and results in the accumulation of myelin debris proximal to the injury site[22]. Astrocytes are capable of phagocytosis of synapses, debris and dead cells in the CNS[4]. To assess whether NogoA labeling on GFAP + reactive astrocytes was due to expression and not a consequence of intracellular myelin debris, we cultured mouse, marmoset and human astrocytes in the absence of myelin. Mouse, marmoset and human astrocytes all expressed NogoA in vitro (Supplementary Fig. 4). Primate astrocytes were observed to be much larger in size when compared to mouse astrocytes as previously demonstrated[23]. Therefore, these findings confirm that expression of NogoA in astrocytes is independent of myelin debris uptake.

**Infiltrating peripheral macrophages in the ischemic zone express NogoA immune-receptor LILRB2.** Following ischemic stroke, the blood–brain barrier (BBB) breaks down, permitting infiltration of blood-borne (peripheral) macrophages into the parenchyma[24]. In parallel, reactive astrocytes can exert inhibitory effects on infiltrating inflammatory cell populations in order to mitigate damage following CNS injury, such as producing anti-inflammatory cytokines and corralling immune infiltrates in a specific region[25,26]. Following ischemic stroke, infiltration of peripheral macrophages peaks around 5–7 DPI in primates, including humans[27,28] (Supplementary Fig. 5a). Consistent with this, we observed that almost all macrophages at 7 DPI in the ischemic zone were of peripheral origin, as indicated by the abundant expression of the macrophage marker Iba1 and absence of the microglia-specific marker TMEM119[29] (Fig. 4a, c), which is in line with previous work in the marmoset[2]. However, we cannot entirely exclude the presence of TMEM119 + microglia in the ischemic zone given *TMEM119* was not annotated in our genome alignment. In tissue, we observed the presence of TMEM119-/

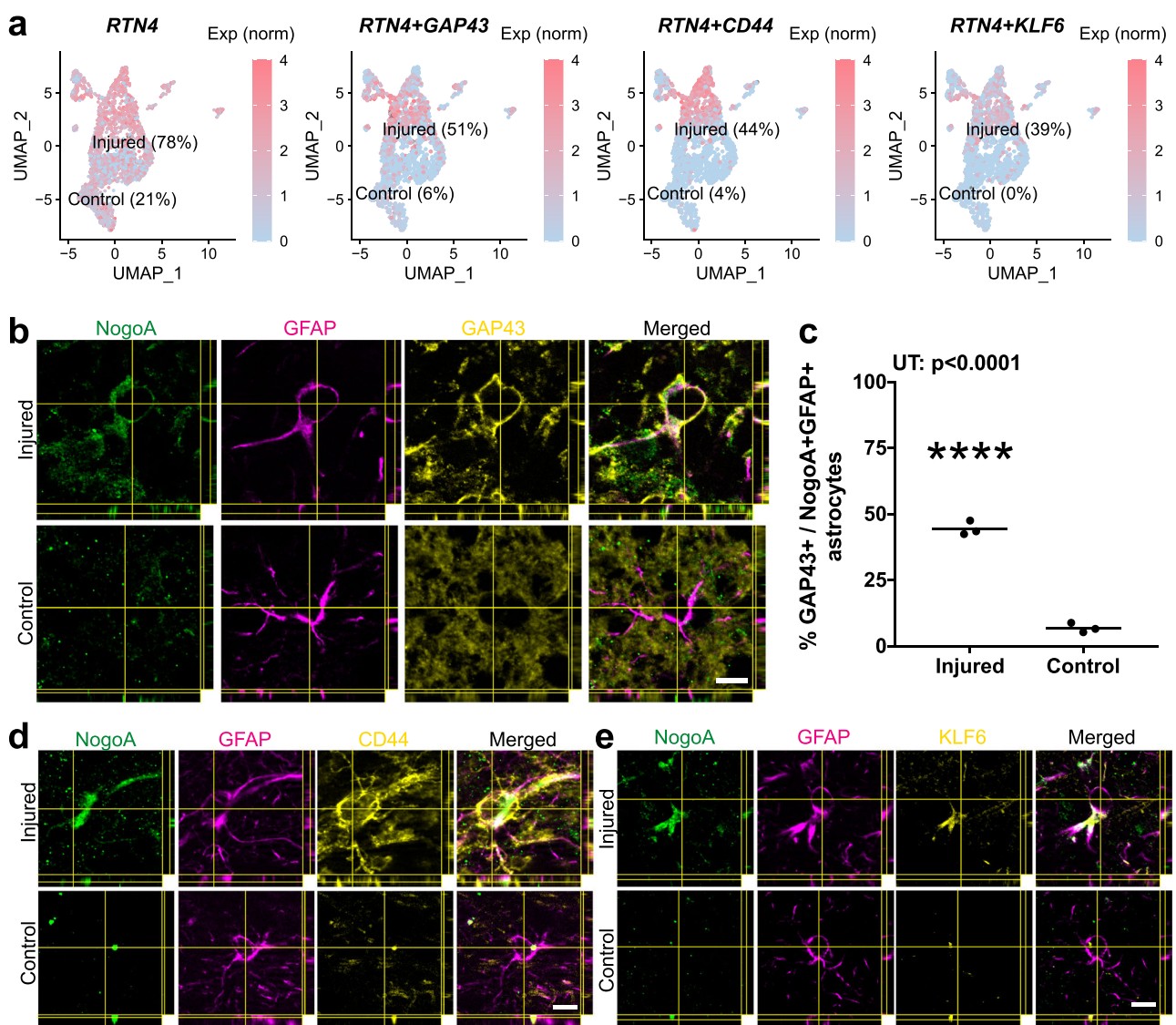

**Fig. 2 In tissue validation of select markers upregulated concomitantly with NogoA in transcriptomic analysis. a** Nuclei-specific expression of *RTN4A*, and concordant *GAP43*, *CD44*, and *KLF6*. **b** Immunohistochemical validation of GAP43 upregulation in NogoA-positive reactive astrocytes from a separate cohort of adult marmosets. Injured = 7 days post-injury (ET-1 injection); Control = non-injected marmoset. **c** Quantification of GAP43 + / NogoA + GFAP + triple-positive astrocytes from (**b**). Each point represents the mean percentage ± SEM from an independent animal ($n = 3$ biological replicates) where at least 40 cells were counted over a minimum of three sections at high magnification with stacks to confirm colocalization of NogoA, GFAP, and GAP43. **d** Validation of CD44 upregulation by immunohistochemistry in marmoset 1-week post-stroke, compared to control ($n = 3$ biological replicates). **e** Validation of KLF6 upregulation by immunohistochemistry in marmoset 1-week post-stroke, compared to control ($n = 3$ biological replicates). *RTN4A* (NogoA): reticulon-4/ neurite outgrowth inhibitor A; GAP43: growth-associated protein 43; KLF6 kruppel-like factor 6, CD44 cell-surface glycoprotein-44; ET-1 endothelin-1; scale bar: 10 μm (**b–d**); UT unpaired *t*-test, statistical test: UT, two-tailed, $t = 19.97$, df = 4, 95% CI; ****$p < 0.0001$. Source data are provided as a Source Data file.

Iba1+ peripheral macrophages within the ischemic zone at 7 DPI (Fig. 4b). By 21 DPI, we observed a significant reduction of peripheral macrophages within the ischemic zone compared to their peak at 7 DPI (Fig. 4c). It is important to note that although other brain resident macrophages, such as perivascular, choroid plexus and meningeal macrophages also present as TMEM119-/ Iba1 + [29], the timing, location, and abundant number of these cells observed at 7 DPI is indicative of a primarily peripheral macrophage population. Therefore, our results indicate that peripheral macrophages infiltrate the ischemic zone parenchyma by 7 DPI, coinciding with the expression of NogoA on reactive astrocytes.

Outside the CNS, the NogoA immune-receptor LILRB2 is primarily expressed on mononuclear phagocytes, B cells, and dendritic cells[30]. Blood-borne macrophages lacking LILRB2 are hyper-adhesive and spread more rapidly, indicating that LILRB2 functionally limits adhesion and cell spreading[31]. While expression of LILRB2 is known on human peripheral monocytes[30], the expression on marmoset monocytes is unknown. Based on these data, we asked whether infiltrating macrophages expressed LILRB2 in marmosets and whether macrophage distribution relative to NogoA+ reactive astrocytes was consistent with a receptor-ligand interaction between cell populations. To address these questions, blood was collected from adult marmosets and mononuclear cells were isolated and cultured. Marmoset macrophages were identified as CD14 + / CD11b + cells[32], and expression of LILRB2 was detected after 7 days in culture, consistent with humans (Fig. 4d). Thus, blood-borne

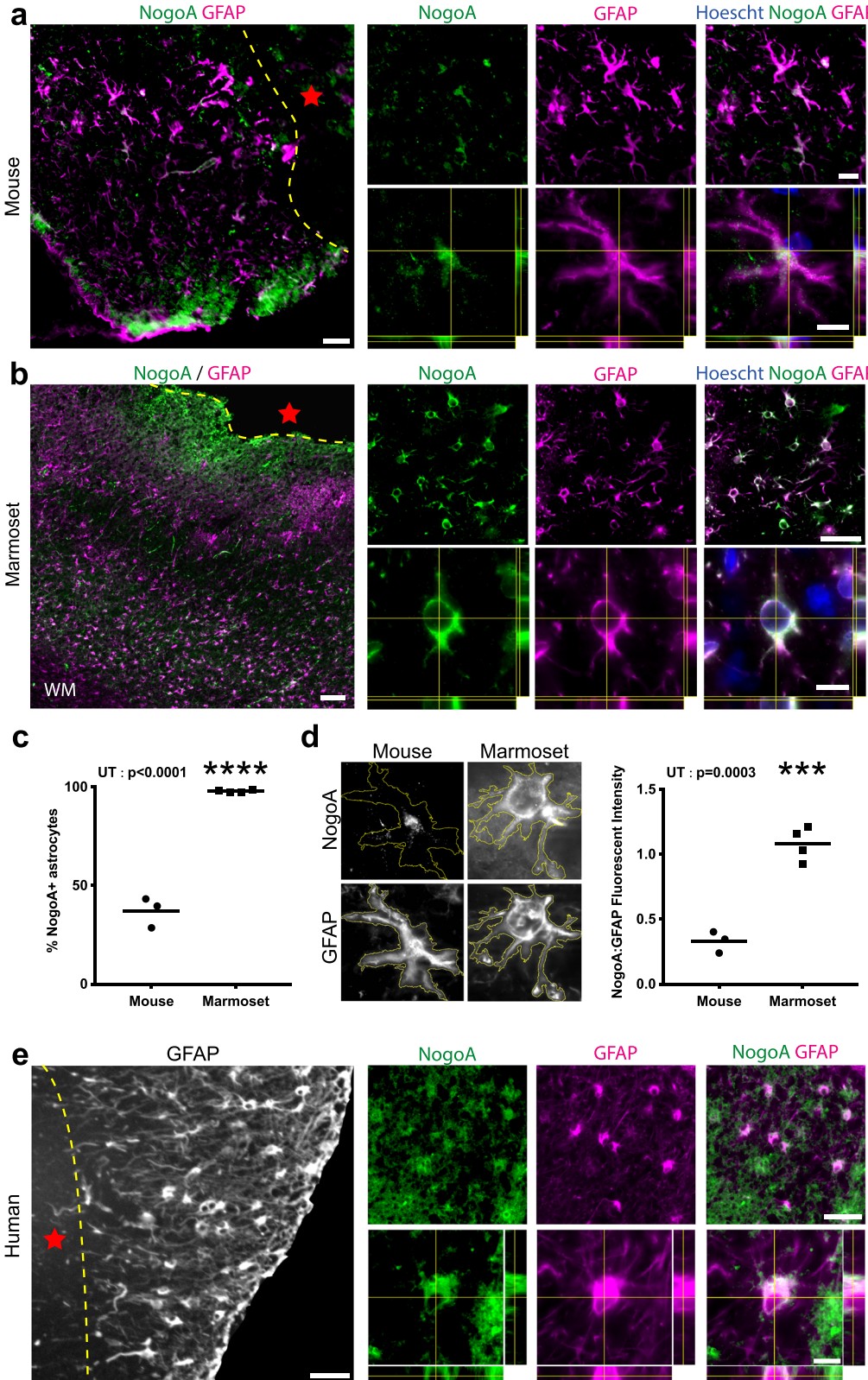

macrophages highly express LILRB2 in the marmoset, as in humans, at baseline.

To determine whether the aforementioned infiltrating peripheral macrophages were expressing LILRB2, we immunolabeled tissues and found a dense distribution of LILRB2+/Iba1+ macrophages in the ischemic zone at 7 DPI (Fig. 4e), compared to controls and other post-ischemic stroke time points

(Supplementary Fig. 5b). The population of LILRB2+/Iba1+ cells exhibited a spherical/ameboid morphology at 7 DPI, indicative of infiltrating peripheral macrophages[33] or phagocytic microglia[34] (Fig. 4e). The LILRB2+/Iba1+ macrophages (Fig. 4e, yellow-hatched line) and TMEM119-/Iba1+ (infiltrating peripheral) macrophages (Fig. 4a, b, yellow-hatched line) were localized to the same region of the ischemic zone, suggesting that they were

**Fig. 3 NogoA expression after ischemic stroke in mouse, marmoset and human. a, b,** and **e** Representative immunofluorescent photomicrographs and stacks with orthogonal views showing NogoA colocalization with GFAP + cells in mouse (**a**) and primate (**b** marmoset; **e** human) cortical tissue adjacent to the ischemic core 3 and 7 days post-ischemic stroke, respectively (**a**: n = 3 biological replicates; **b**: n = 4 biological replicates; **e**: n = 1 biological replicate). **c** Quantification of NogoA + /GFAP + cells in mouse versus marmoset expressed as a percentage of total counted astrocytes. Each point represents the mean percentage ± SEM from an individual animal (mouse: n = 3 biological replicates, marmoset: n = 4 biological replicates), where at least 30 cells were counted over a minimum of three sections at high magnification with stacks to confirm colocalization of NogoA and GFAP (**d**) Quantification of NogoA fluorescent intensity, normalized against GFAP, in mouse versus marmoset. Each point represents the mean fluorescent intensity ± SEM of NogoA:GFAP for an individual animal (mouse: n = 3 biological replicates, marmoset: n = 4 biological replicates) where at least ten cells from previous counts in (**c**) were analyzed at high magnification. NogoA: neurite outgrowth inhibitor A; GFAP glial fibrillary acidic protein, hoechst: nuclear stain; WM white matter, yellow-hatched line: border of ischemic core; red star: ischemic core; scale bars: 100 µm (**a, b,** and **e**: left), 50 µm (**a, b,** and **e**: top), 10 µm (**a, b,** and **e**: bottom); UT unpaired t-test; statistical test: UT, p-value < 0.0001; two-tailed, t = 16.38, df = 5, CI = 95% (**c**), UT, p-value = 0.0003, two-tailed, t = 8.764, df = 5, CI = 95% (**d**); ***p < 0.001; ****p < 0.0001. Source data are provided as a Source Data file.

the same population of cells. Furthermore, at 7 DPI we observed that deeper infiltrating Iba1+ macrophages had reduced LILRB2 expression and exhibited ramified morphologies compared to the macrophages more proximal to the cortical surface, which highly expressed LILRB2 and were spherical/ ameboid in morphology (Supplementary Fig. 5c). At 7 DPI, the spatial distribution of LILRB2 and NogoA expression were juxtaposed at the ischemic zone (Fig. 4f). These results demonstrate that LILRB2 is upregulated on a population of Iba1+ peripheral macrophages 7 DPI with complementary expression to NogoA upregulation on GFAP + astrocytes. Combined with our transcriptomic analysis of *RTN4A* (NogoA) astrocyte function and indication of involvement in regulating leukocyte trafficking, these data suggest a potential ligand-receptor interaction contributing to astrocytic corralling of macrophages.

**NogoA-positive reactive astrocytes induce macrophage repulsion through two functional domains.** Genetic deletion of the mouse homolog of LILRB2, PirB, after stroke results in an attenuated reactive astrocyte response[35], indicating a molecular role for LILRB2 related to astrocytes after injury. Additionally, NogoA signaling has been previously demonstrated to limit the migration of microglia in vitro[36]. NogoA-LILRB2 interaction typically leads to downstream POSH/ Shroom3 and/ or RhoA/ ROCK-dependent actin cytoskeletal reorganization[37], ultimately resulting in neurite collapse or cell repulsion. This knowledge, combined with our transcriptomic analysis of *RTN4A* (NogoA)+ astrocyte function, prompted us to consider a NogoA/ LILRB2-dependent immune regulatory function following CNS injury. The function of NogoA/ LILRB2 signaling on peripheral macrophages was investigated using a human monocyte cell line, THP-1-derived macrophages. Analysis of previous transcriptomic data from Gosselin, Skola[29] revealed the enrichment of LILRB2 on human monocytes (Fig. 5a), with an absence of other NogoA binding partners, such as NgR1 and S1PR2. We confirmed this in THP-1-derived macrophages (Fig. 5b). LILRB2 + THP-1-derived macrophages, co-expressing Iba1, were phagocytic and exhibited expected morphologies (Fig. 5c), including but not limited to multi-nucleation, granulation and heterogeneity in cell shape and size, as described in Daigneault, Preston[38]. We subsequently investigated if the two potent functional domains of NogoA (Nogo-66 and Nogo-Δ20) (Fig. 5d), could activate LILRB2-signaling on THP-1-derived macrophages by examining key downstream effectors: POSH, Shroom3, RhoA, and ROCK1. Nogo-66 treatment induced significant elevation of LILRB2 and POSH at 120 min (Fig. 5e, g). Shroom3 and RhoA elevation were detected more acutely at 60 and 30 min, respectively, returning to control levels at subsequent time points analyzed (Fig. 5e, g). Nogo-Δ20 treatment induced significant elevation of LILRB2 at 60 min, which was sustained up to 120 min (Fig. 5f, h). POSH

elevation was detected more acutely starting at 10 min and remaining elevated (Fig. 5f, h). Shroom3 and RhoA were elevated at 120 min (Fig. 5f, h). No significant changes in ROCK1 expression levels were observed up to 120 min of stimulation with Nogo-66 or Nogo-Δ20 (Fig. 5e–h). These data suggest that NogoA induces LILRB2 signaling through the POSH/ Shroom3-dependent pathway in human macrophages via its two distinct functional domains either directly or indirectly. Finally, stripe assays were performed to demonstrate a functional role for NogoA/ LILRB2 signaling in THP-1 human macrophages. Stimulation with either Nogo-66 or Nogo-Δ20 revealed a significantly greater number of Iba1+ macrophages between immobilized human NogoA (Nogo-66-Fc or Nogo-Δ20-Fc) stripes compared to immobilized control-Fc stripes (Fig. 5i, j). Our results demonstrate that LILRB2 blockade significantly attenuates the effect of NogoA signaling on human macrophages resulting in a macrophage distribution indistinguishable from controls (Fig. 5i, j). These data demonstrate that NogoA/ LILRB2 signaling leads to macrophage repulsion. Taken together, these data provide substantive evidence that NogoA expression on GFAP + reactive astrocytes contributes to the corralling of LILRB2 + infiltrating monocytes and macrophages in the primate brain post-ischemic stroke.

Although Nogo-Δ20 is not able to pull down LILRB2 from human macrophage cell lysates (Fig. 5k), blocking LILRB2 overcomes Nogo-Δ20-mediated repulsion in vitro (Fig. 5j). Nogo-Δ20 has not been demonstrated to signal through LILRB2 directly, but rather various integrins[39] (Fig. 5d). As a negative regulator of integrins, the absence or blockade of PirB in macrophages can cause enhanced integrin signaling, resulting in hyper-adhesive macrophages[31]. We also considered that Nogo-Δ20 effects may occur via S1PR2 signaling. S1PR2 is an established Nogo-Δ20 receptor (Fig. 5d). Very little is known about the detailed composition of the NogoA receptor complexes, and there is the possibility that different subunits may form a multisubunit heterodimeric complex akin to neurotrophin or Wnt receptors. Thus, antibodies against one subunit, such as LILRB2, could sterically hinder the binding of other subunits to a different ligand site, such as S1PR2. However, given the absence of S1PR2 on human macrophages (Fig. 5b), we hypothesize that blocking LILRB2 in our experimental paradigm activated integrin-mediated signaling which overcame Nogo-Δ20-mediated repulsion. This hypothesis is supported by evidence demonstrating that activating integrin-β1 could overcome Nogo-Δ20-mediated repulsion[39], and our own experiments demonstrating that integrin-α4 and -β3 are significantly upregulated on human macrophages following treatment with a LILRB2 blocking antibody (Fig. 5l). Thus, these data demonstrate that reactive astrocyte-mediated repulsion of infiltrating macrophages between 7–14 DPI occurs through NogoA-LILRB2 signaling in the primate.

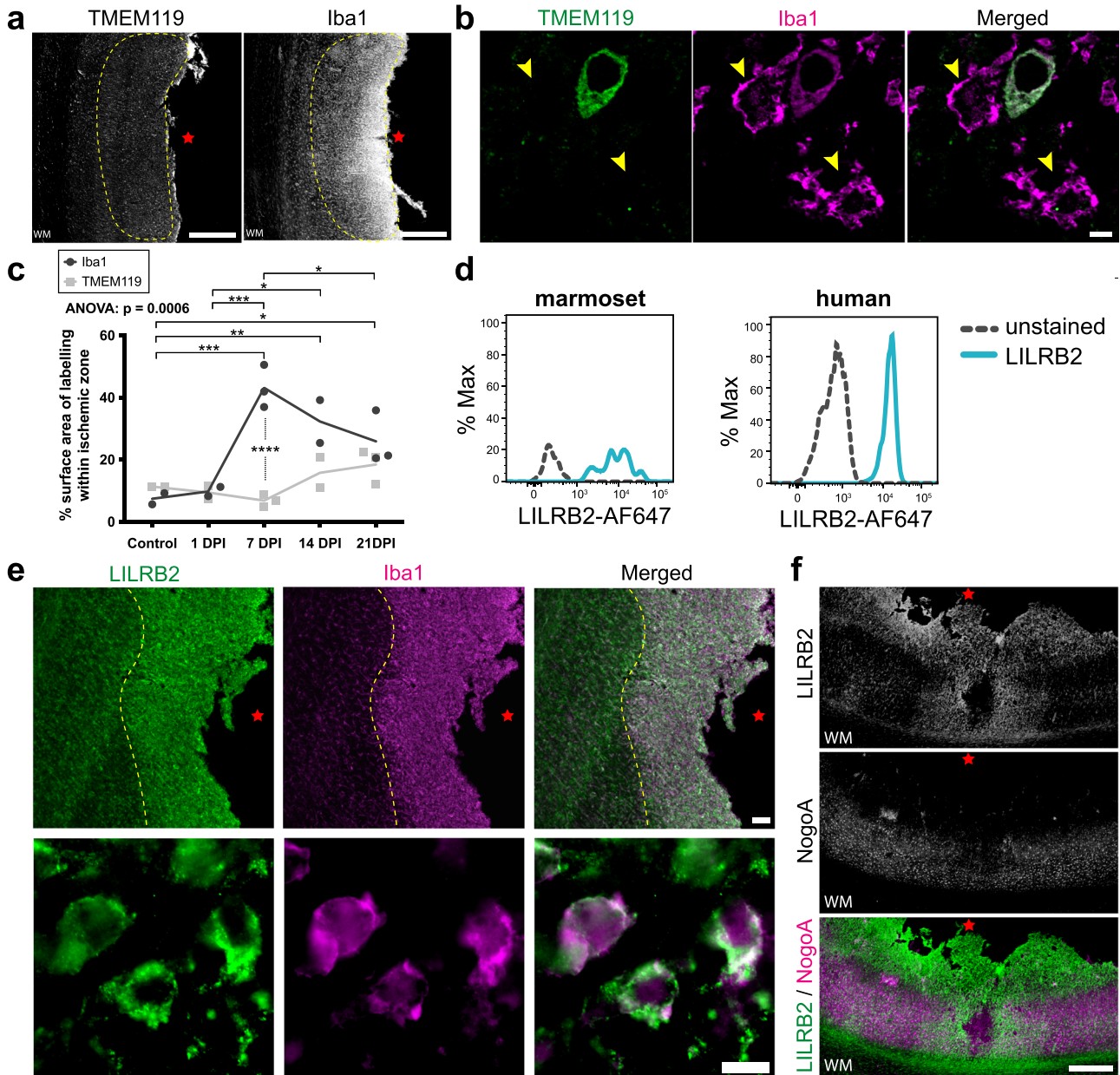

**Fig. 4 Peripheral macrophage infiltration in the marmoset post-ischemic stroke. a** Representative low-power photomicrographs revealing the low TMEM119 (microglia-specific) and high Iba1 (macrophages) expression within the ischemic zone of marmoset 1-week post-ischemic stroke cortical tissue by DAB immunohistochemistry ($n = 3$ biological replicates). **b** High-power photomicrographs identify a lack of TMEM119 colocalization with Iba1+ cells in the ischemic zone of 1-week post-ischemic stroke marmoset ($n = 3$ biological replicates). **c** Quantification of peripheral macrophage infiltration in the marmoset post-ischemic stroke. Each point represents the mean surface area labeled with Iba1 versus TMEM119, calculated from a minimum of three sections per marmoset across various post-ischemic stroke recovery time points ($n = 2$ biological replicates: control, 1 DPI and 14 DPI; $n = 3$ biological replicates: 7 DPI and 21 DPI). **d** Marmoset and human macrophage analysis. Fluorescent-activated cell sorting data depicting NogoA immune-receptor (LILRB2) expression in marmoset and human CD14+/CD11b+ macrophage populations. **e** Representative low and high-power photomicrographs demonstrating LILRB2 colocalization with Iba1+ cells in the ischemic zone of 1-week post-ischemic stroke marmoset ($n = 3$ biological replicates). **f** Representative pseudo colored LILRB2 and NogoA photomicrographs overlaid to show spatial distribution of labeling ($n = 3$ biological replicates). TMEM119: transmembrane protein 119; Iba1: ionized calcium-binding adapter molecule 1; WM white matter; yellow hatched line: cell population of interest with lack of TMEM119 expression; red star: ischemic core; yellow arrowheads: TMEM119-/Iba1+ peripheral macrophages; DPI days post-ischemic stroke, LILRB2 leukocyte immunoglobulin-like receptor B2, NogoA, neurite outgrowth inhibitor A; scale bars: 500 μm (**a**, **f**), 5 μm (**b**), 100 μm (**e**: top), 10 μm (**e**: bottom); statistical tests: ordinary two way-ANOVA with Sidak's multiple comparisons; ANOVA $p$-value = 0.0006; *$p < 0.05$; **$p < 0.01$; ***$p < 0.001$. Source data and tabulated statistics for (**c**) are provided as a Source Data file.

## Discussion

Here we investigated the transcriptomic changes in NHP astrocytes after brain injury at single-nuclei resolution. Deep molecular profiling revealed a remarkable shift in astrocyte gene expression

following ischemic stroke in the marmoset monkey. We observed significant upregulation of genes contributing to immunomodulatory signaling pathways in the post-stroke brain. Notably, within the top 30 DEGs alone, we identified that *RTN4A*

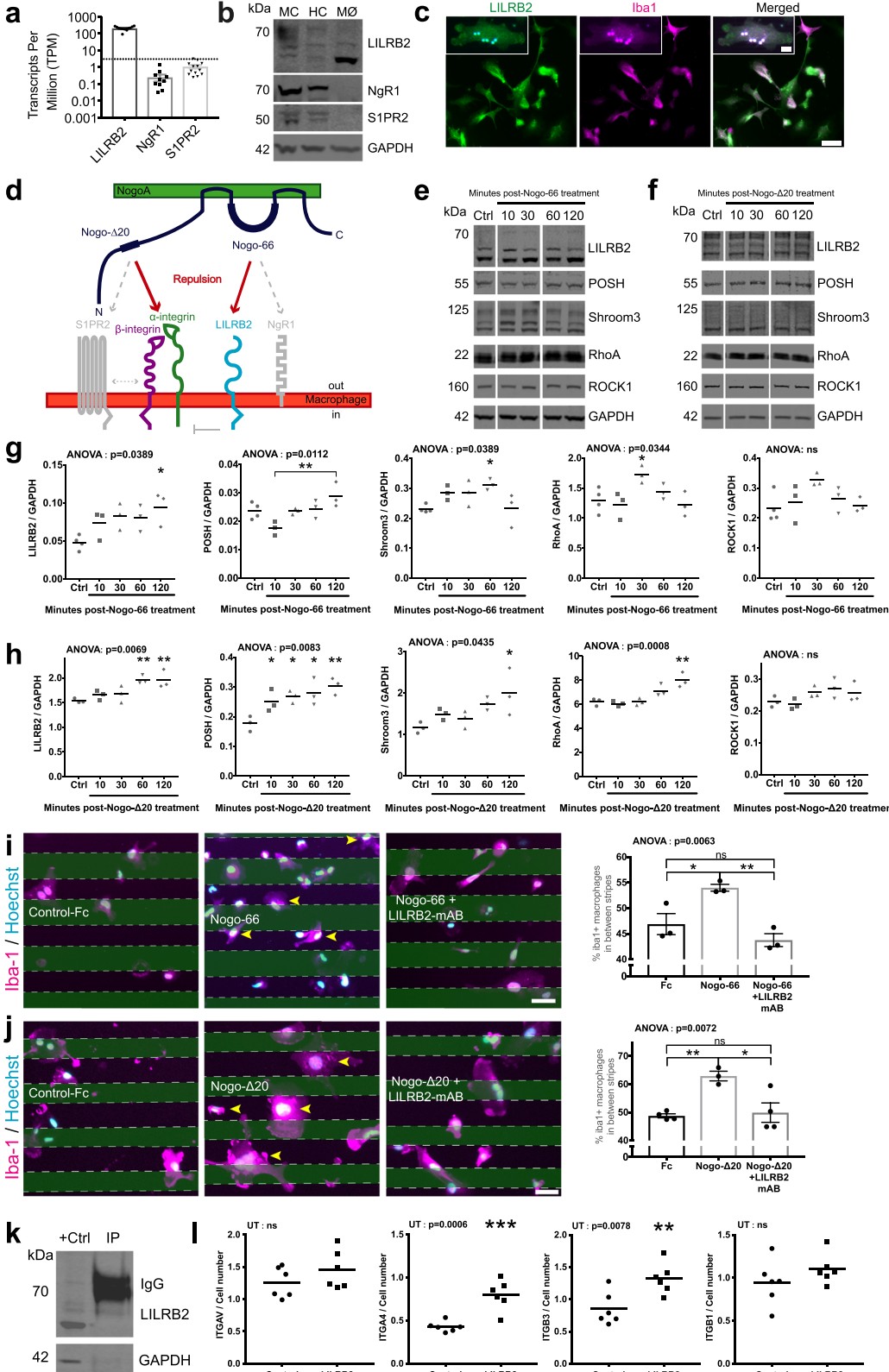

(NogoA), a membrane protein previously described as an oligodendrocyte-associated neurite outgrowth inhibitor, is also expressed by astrocytes and its expression is transiently upregulated by reactive astrocytes adjacent to the ischemic core, including in the human cortex. Finally, we demonstrated that NogoA likely functions in these reactive astrocytes to corral peripheral macrophages from spreading into healthy adjacent CNS tissue. Our data highlight differences in astrocyte identity across model organisms and emphasizes the need for models that better recapitulate primate, including human, phenomena for the development of therapeutic treatments.

To date, the most widely accepted role for reactive astrocytes is to create a glial scar, as a structural and molecular barrier to repair and regeneration[40,41]. However, astrocyte reactivity is not

**Fig. 5 NogoA/ LILRB2 signaling induces repulsion of human macrophages. a** Human RNA-seq data from Gosselin, Skola[29] demonstrating known NogoA ligand binding partners in transcripts per million detected in human monocytes. **b** Qualitative immunoblot for LILRB2, NgR1 and S1PR2 in marmoset (MC) and human (HC) cortical tissues, and THP-1-derived human macrophages (MØ) (*n* = 1 biological replicate/tissue as a qualitative validation of published data from (**a**)). **c** LILRB2/ Iba1 immunostaining of THP-1-derived human macrophages, in vitro. THP-1-derived human macrophages were tested for phagocytosis of latex beads (c, magnified box) (*n* = 2 biological replicates over at least three serial dilutions). **d** Schematic depicting NogoA domains and receptors. **e, f** Representative immunoblots for THP-1-derived human macrophages treated with human Nogo-66 (**e**) or Nogo-Δ20 (**f**) for 0, 10, 30, 60, and 120 m. **g, h** Scatter plots depict densitometric quantification of protein normalized to GAPDH for **e** and **f**, respectively. Each point represents the mean of 2 technical replicates for independent THP-1-derived macrophage cohorts (*n* = 3 biological replicates/ condition, excluding PBS control for **e** and **g**, where *n* = 4 biological replicates/condition, with no technical replicates). **i, j** Representative immunostaining of the in vitro stripe (repulsion) assay for human Control-Fc, NogoA (Nogo-66-Fc or Nogo-Δ20-Fc) and NogoA with LILRB2 blocker pre-treatment of macrophages. The scatter plots with bars depict the mean percentage ± SEM of Iba1 + cells situated in between stripes (*n* = 3 biological replicates/ condition, at least three images per condition). **k** Nogo-Δ20 pull down assay of LILRB2 from THP-1 macrophage cell lysate (*n* = 1 biological replicate in triplicate). **l** Effect of LILRB2 treatment on THP-1-derived macrophage integrin expression. Scatter plots depict densitometric quantification of protein normalized to cell number. Each point represents the signal independent THP-1-derived macrophage cohort (*n* = 6 biological replicates/condition over one experiment). LILRB2 leukocyte immunoglobulin-like receptor B2, NgR1 Nogo Receptor 1, S1PR2 sphingosine-1-phosphate receptor 2; GAPDH: glyceraldehyde 3-phosphate dehydrogenase; kDa: kilodalton; MC: marmoset cortical tissue, HC human cortical tissue, MØ macrophages, Iba1 ionized calcium-binding adapter molecule 1, POSH plenty of sarcoma homology domain 3, ROCK1 Rho kinase 1, NogoA neurite outgrowth inhibitor A, mAB monoclonal antibody, +Ctrl positive control, IP immunoprecipitation, IgG immunoglobulin, ITGAV integrin subunit alpha V, ITGA4 integrin subunit alpha 4, ITGB3 integrin subunit beta 3, ITGB1 integrin subunit beta 1, yellow arrowheads: NogoA-repelled cells (**i, j**); scale bars: 50 μm (**c, i,** and **j**), 10 μm (**c,** magnified box); statistical tests: ordinary one way-ANOVA with post-hoc Dunnett's OR Tukey's multiple comparisons (**g–j**), unpaired *T*-test (UT) (l); *\*p* < 0.05; *\*\*p* < 0.01; *\*\*\*p* < 0.001; ns: not significant. Source data and tabulated statistics for (**g, h, i, j,** and **l**) are provided as a Source Data file.

necessarily synonymous with repair inhibition, since reactive astrocytes provide crucial neuroprotection after CNS injuries, e.g.: trophic support[42–44], attenuation of excitotoxicity[11,45–47], elimination of cell debris[48], and regulation of neuroinflammation[26,49]. In line with this latter function, we observed significant upregulation of genes involved in regulation of interferon-gamma signaling, cytokine production, response to cytokines, and defense response to virus. Additionally, given the post-stroke time point of 1-week, which coincides with the peak of peripheral immune infiltration[27,28], it is not surprising that within the top 30 DEGs the majority were genes associated with immunomodulatory functions. However, the reality that reactive astrocyte characteristics significantly differ between rodents and humans, including diverging microanatomical[50] and transcriptional characteristics[51] that collectively underpin functional differences, may explain the failure to observe NogoA on reactive astrocytes over the past decades. A recent study directly comparing primary resting astrocytes derived from mouse versus human brain, demonstrated that more than 600 genes enriched in human astrocytes were not similarly enriched in mouse astrocytes[51]. Furthermore, extrinsic factors that influence astrocyte gene expression and subsequent function, such as the immune response, differ significantly between rodents and primates. Mouse and human macrophages can also respond very differently to the same stimuli[52]; for example, lipopolysaccharide induces upregulation of iNOS in mouse macrophages but upregulation of CCL20, CXCL13, IL-7R, P2RX7, and STAT4 in human macrophages[53]. The combination of these interspecies differences is likely to culminate in variability in additional mechanisms associated with the interplay between astrocytes and monocytes/ macrophages after CNS injury. Although we observed low levels of NogoA expression on reactive astrocytes in mouse, NogoA is not upregulated at 3 days post-ischemic stroke[19,20], equivalent to 1-week post stroke in marmoset, at which time there is an influx of blood-borne macrophages. As such, the enrichment of NogoA on a large proportion of reactive astrocytes in primates, including human, but not rodents[19,20], which coincides temporally with peripheral macrophage infiltration after ischemic stroke, is likely a consequence of requiring a more complex reactive astrocyte response to a more complex immune response. The potential addition of NogoA to astrocyte-mediated corralling of infiltrating leukocytes reinforces the physiological importance of this

neuroprotective mechanism to spare healthy brain tissue and confine the injury to a discrete region.

There are some limitations to the current study that merit discussion. First, although we have explored various time points immunohistochemically, our transcriptomic dataset is limited to the 1-week time point after ischemic brain injury in the marmoset. Transcriptomic analyses of astroglia at additional time points may provide greater clarity of the temporal picture of reactive astrocyte function beyond the acute time points. Second, we have restricted our analysis to regions proximal to the lesion site. We have not explored whether there are transcriptomic and phenotypic changes in astrocytes in regions distal to the injury site. Future studies documenting the transcriptomic and phenotypic status of astrocytes in regions distal to the injury site would provide a more complete spatial picture of the astroglial response. It is conceivable, for example, that the presence of extensive interhemispheric projections result in changes in astrocytes in the contralateral hemisphere since denervation is a well-recognized trigger of astrocyte reactivity and changes in gene expression. Third, our functional analyses focus on a particular protein-coding gene of interest. Given the extensive list of differentially expressed immunomodulatory genes in astrocytes, there are likely to be other gene products that influence reactive astrocyte function after brain injury in primates. Despite these limitations, the current study advances our understanding of reactive astrocyte function, revealing that there are important differences between rodent and primate, including human, after ischemic brain injury which have significant clinical implications for current and future therapeutics.

With respect to therapeutic intervention, NogoA and other myelin-associated inhibitors have been extensively studied in their capacity as inhibitors of repair and have been identified as promising targets for therapeutic intervention, especially for the treatment of SCI[54–63] and stroke[64–70]. Various neutralizing antibodies and competitive inhibitors against NogoA and receptor bodies, as well as genetic deletions, have been used to demonstrate enhanced sprouting and regeneration after CNS lesions[22]. One of these, a NogoA neutralizing antibody, IN-1, has been tested in rodent models of SCI[59–61,71], followed by NHPs[55,72,73], and finally human subjects[74]. The Phase I clinical trial using ATI335 (NogoA neutralizing antibody) in acute paraplegic and tetraplegic patients reported that it is well

tolerated following intrathecal administration[74] and has entered Phase II trials in subacute SCI (ClinicalTrials.gov NCT03935321). In addition, a soluble NgR decoy blocking Nogo, MAG and OMgp ligands is efficacious preclinically in chronic SCI[75,76], and has entered clinical trials for that indication (ClinicalTrials.gov NCT03989440). The current study provides evidence that consideration of timing of the specific intervention is essential to ensure appropriate therapeutic benefit.

Importantly, our findings from acute to subacute time points following ischemic stroke in the marmoset indicate that NogoA antagonism might restrict astrocyte-mediated corralling of infiltrating macrophages. In addition, NogoA blockade might have indirect immunomodulatory effects at early time points. Here, we find that NogoA+ astrocytes are enriched with genes involved in leukocyte trafficking, which accurately recapitulates our in vivo and in vitro observations of peripheral macrophage corralling by this cell population. We know that reactive astrocytes can physically restrict migration of leukocytes, including macrophages, after CNS injury to limit their infiltration into adjacent healthy tissue[26,77]. While immune infiltrates are crucial for the clearance of detrimental cell and myelin debris[78], the inflammatory consequences may exacerbate secondary neuronal cell death. For example, if proliferating or corralling reactive astrocytes are ablated following CNS trauma, the result is propagation of leukocyte infiltration, severe demyelination, neuronal and oligodendrocyte cell death and pronounced motor deficits[79,80]. Expression of the receptors ERα (ESR1), GP130 (IL6ST), DRD2, and C5aR1[81–85] in rodent reactive astrocytes provides potential alternative mechanisms to limit immune infiltration. However, DRD2 and C5aR1 are absent or negligible in marmoset reactive astrocytes with only ESR1 & IL6ST significantly upregulated 1-week after stroke. In primates, NogoA signaling may be crucial to fully restrict macrophage infiltration into surrounding healthy CNS parenchyma after stroke. Therefore, different mechanisms may mediate macrophage corralling in rodents and primates. NogoA-directed therapies in primate may titrate both early modulation of inflammation, as well as longer term regulation of axonal sprouting and neural repair.

In sum, our results support the notion that a NHP ischemic stroke model is more representative of human sequelae than the rodent, highlighted by our identification of abundant NogoA+ reactive astrocytes both in marmoset and human cortical tissue 1-week post-stroke. Given the complex and species-specific response of astroglial cells to CNS injury, it is essential to complete a detailed molecular characterization of NHP preclinical models, considering the immune system implication amongst others, to advance ischemic stroke research and therapeutic development.

## Methods

**Animals.** Adult common marmosets (*Callithrix jacchus* > 18 months; $n = 15$ fixed, $n = 16$ fresh, $n = 6$ for transcriptomics) were used in this study. Animals were subdivided into uninjured control and injured cohorts, comprising 1, 7, 14, and 21 days post-injury (DPI) recovery periods (fixed: $n = 3$ per time point; fresh: $n = 3$ per time point, excluding 14 DPI, where $n = 4$). $n = 3$ for transcriptomics were allocated to the 7 DPI recovery period and $n = 3$ uninjured marmosets were used as control. Experiments were conducted according to the Australian Code of Practice for the Care and Use of Animals for Scientific Purposes and were approved by the Monash University Animal Ethics Committee. Animals were obtained and housed at the National Nonhuman Primate Breeding and Research Facility (Monash University).

**ET-1-induced focal ischemic stroke.** Preoperative procedures, anesthesia and surgery were all performed on adult marmoset monkeys[2]. In short, anesthesia was induced using Alfaxalone (5 mg/kg) and maintained using inspired isoflurane (0.5-4%) throughout all surgical procedures. Induction of focal ischemic injury to adult marmoset primary visual cortex (V1) was achieved by vasoconstrictor-mediated vascular occlusion of the calcarine branch of the posterior cerebral artery, which

supplies operculum V1. Following midline incision, craniotomy, and dural resection, intracortical injections of endothelin-1 (ET-1: 1 mg/mL; rate: 0.1 μL/30 s pulse + 30 s intervals, totaling 0.7 μL over seven sites) proximal to the PCAca were performed. Upon completion of injections, the craniotomy was replaced and positioned with tissue adhesive (Vetbond 3 M) and the skin sutured closed. Uninjured animals were used as controls. Stroke model is summarized in Fig. 1a and Supplementary Fig. 1a.

**Single nuclei 10x genomic sequencing and analysis.** Adult marmosets in both injured (1-week post-ET-1-induced ischemic stroke, $n = 3$) and uninjured cohorts (control, $n = 3$) were administered an overdose of sodium pentobarbitone (100 mg/kg; IM). Following the loss of corneal and muscular reflexes, the animals were decapitated, and the brain removed under aseptic conditions. Cerebral tissues were rinsed in chilled PBS to remove excess blood, before microdissection of regions of interest within V1, including operculum V1 but excluding calcarine V1 and associated white matter (Fig. 1a). In stroke animals, this also included the core and penumbra. Dissected tissues were placed in sterile tubes and dropped into liquid nitrogen before storage at −80°. The procedures/ dissections were performed in chilled RNAse-free PBS with RNase-free sterilized instruments under RNase-free conditions. Approximate time from apnea to snap-freezing ranged from 20–30 m. All six samples passed QC.

**Nuclei Isolation.** All buffers were ice-cold and all reagents used for consequent nuclear isolation were molecular biology grade unless stated otherwise.

Frozen cerebral tissues from the injured cohort were finely pulverized to powder in liquid nitrogen with mortar and pestle (Coorstek #60316, #60317). 50 mg of pulverized tissue was added into 5 mL of ice-cold lysis buffer: 320 mM sucrose (Sigma #S0389), 5 mM CaCl₂ (Sigma #21115), 3 mM Mg(Ace)₂ (Sigma #63052), 10 mM Tris-HCl (pH 8) (AmericanBio #AB14043), protease inhibitors w/o EDTA (Roche #11836170001), 0.1 mM EDTA (AmericanBio #AB00502), RNAse inhibitor (80 U/mL) (Roche #03335402001), 1 mM DTT (Sigma #43186), 0.1% TX-100 (v/v) (Sigma #T8787). Reagents: DTT, RNAse Protector, protease inhibitors, TX-100 were added immediately before use. The suspension was transferred to Dounce tissue grinder (15 mL volume, Wheaton #357544; autoclaved, RNAse free, ice-cold) and homogenized with loose and tight pestles, 30 cycles each, with constant pressure and without introduction of air. The homogenate was strained through 40 um tube top cell strainer (Corning #352340) which was pre-wetted with 1 mL isolation buffer: 1800 mM sucrose (Sigma #S0389), 3 mM Mg(Ace)₂ (Sigma #63052), 10 mM Tris-HCl (pH 8) (AmericanBio #AB14043), protease inhibitors w/o EDTA (Roche #11836170001), RNAse inhibitor (80 U/mL) (Roche #03335402001), 1 mM DTT (Sigma #43186). Additional 9 mL of isolation buffer was added to wash the strainer. Final 15 mL of solution was mixed by inverting the tube 10x and carefully pipetted into two ultracentrifuge tubes (Beckman Coulter #344059) onto the isolation buffer cushion (5 mL) without disrupting the phases. The tubes were centrifuged at $30000 \times g$, for 60 min at 4 °C on ultracentrifuge (Beckman L7-65) and rotor (Beckman SW41-Ti). Upon end of ultracentrifugation, the supernatant was carefully and completely removed and 100 ul of resuspension buffer (250 mM sucrose (Sigma #S0389), 25 mM KCl (Sigma #60142), 5 mM MgCl₂ (Sigma #M1028), 20 mM Tris-HCl (pH 7.5) (AmericanBio #AB14043; Sigma #T2413), protease inhibitors w/o EDTA (Roche #11836170001), RNAse inhibitor (80 U/mL) (Roche #03335402001), 1 mM DTT (Sigma #43186)) was added dropwise on the pellet in each tube and incubated on ice for 15 min. Pellets were gently dissolved by pipetting 30x with 1 mL pipette tip, pooled and filtered through 40 um tube top cell strainer (Corning #352340). Finally, nuclei were counted on hemocytometer and diluted to 1 million/mL with sample-run buffer: 0.1% BSA (Gemini Bio-Products #700-106 P), RNAse inhibitor (80 U/mL) (Roche #03335402001), 1 mM DTT (Sigma #43186) in DPBS (Gibco #14190).

Frozen cerebral tissues from the control cohort were homogenized using a glass Dounce tissue grinder (2 mL volume, Sigma, Kimble #D8938-1SET) with loose and tight pestles, 25 cycles each, in 2 mL ice-cold lysis buffer (Nuclei PURE Lysis buffer (Sigma #NUC201-1KT) + 0.1% NP40 (Invitrogen #FNN0021) + RNAse inhibitor (Invitrogen #10777-019). Following homogenization, an additional 2 mL of ice-cold lysis buffer was added and the homogenates were incubated on ice for 5 min. The homogenates were then transferred to 15 mL tubes and centrifuged at $500 \times g$ for 5 min at 4 °C. The supernatant was subsequently removed and discarded. The remaining pellets were then washed with 3 mL ice-cold lysis buffer and incubated on ice for 5 min. After another round of centrifugation, the nuclei were washed in 2 mL nuclei resuspension buffer (NSB: Dulbecco's phosphate-buffered saline + 1% BSA (Ambion #AM2616) + 1x Proteinase inhibitor (Roche #11836153001) + 0.2 U/μL RNAse inhibitor (Invitrogen #10777-019) and centrifuged again for another 5 min with the same parameters. Isolated nuclei were resuspended in 500 μL NSB or approximately 5 million/400 μL, factoring in 30% cell loss from centrifugation. Nuclei were then filtered through a 30 μm cell strainer and counted with Trypan blue using a hemocytometer. Nuclei were subsequently stained with Hoechst 333258 at 1:1000, then processed through FACs analysis/ sorting (Influx3, 70 μm nozzle, PSI 20; with nuclei present in approximately 5–20% of events). DAPI + nuclei were then sorted into a collection tube containing NSB with RNAse inhibitor (Invitrogen #10777-019) at a concentration of 1000 nuclei/μL.

**Single cell/nucleus microfluidic capture, cDNA synthesis and RNAseq library preparation**. The nuclei from the injured cohort were placed on ice and taken to Yale Center for Genome Analysis core facility, and the nuclei from the control cohort were placed on ice and taken to Micromon Genomics facility, each for single nucleus RNA sequencing. Samples were processed within 15 min with targeted nuclei recovery of 10,000 nuclei per sample. Nuclei suspensions were packaged into single-nuclei beads and the cDNA libraries were constructed using Chromium Single Cell 3' v3 Chemistry (10x Genomics) on microfluidic Chromium System (10x Genomics) by following precisely manufacturers detailed protocol (CG000183_ChromiumSingleCell3'_v3_UG_RevA). Specifically, due to limitations imposed by source RNA quantity, cDNA from nuclei was amplified for 8–14 cycles.

**Sequencing of libraries**. In order to reach optimal sequencing depth (25,000 raw reads per nucleus), single cell and/or nucleus libraries were run using paired-end sequencing with single indexing on the HiSeq 4000 platform (Illumina) by following manufacturer's instructions (Illumina, 10x Genomics) (CG000183_ChromiumSingleCell3'_v3_UG_RevA). To avoid lane bias, multiple uniquely indexed samples were mixed and distributed over several lanes.

Similarly, sequencing of the control cohort aimed to reach a sequencing depth of 25,000 raw reads per nucleus by running single-end sequencing of 98b reads in two sequencing lanes (FCL) for at least 400 m raw reads per lane using MGITech MGISEQ2000RS platform via MGIEasy V3 chemistry.

**Cell de-multiplexing, reads alignment and gene expression quantification**. First, the *cellranger mkfastq*[86] was used to demultiplex the raw base call (BCL) files obtained from Illumina HiSeq 4000 sequencers into FASTQ files. Second, the *cellranger mkref* was used to build a reference index for the marmoset reference genome (calJac3), and the GTF format Ensembl[87] gene annotation (Callithrix_jacchus.C_jacchus3.2.1.91.gtf) was processed to the transcriptional model specified for single nuclei sequencing. Third, the *cellranger count* was used to perform read alignments to the reference genome, cell filtering and counting, and gene UMI quantification for every single cell. Notably, all parameters of *cellranger* were set as default, except for "–expect-cells = 10,000".

**Removal of nuclei doublets**. For the injured cohort, nuclei doublets or clumps were removed through post-transcriptomic analysis, using *Scrublet*[88] software with a filtered feature barcode matrix (i.e., *matrix.mtx*) generated by *cellranger* as input. To optimize the analysis, we customized some parameters based on in-house computing trials, i.e., *expected_doublet_rate = 0.05, min_counts = 2, min_cells = 3, min_gene_variability_pctl = 85, n_prin_comps = 30*.

For the control cohort, Seurat's standard pre-processing and quality control workflow was followed to remove low-quality cells, empty droplets, cell doublets, and multiplets[89]. To optimize the analysis, the QC metrics were customized (200 > nFeature_RNA < 2500) based on the distribution of features-counts.

**Data analysis and mining**. During the first stages of analysis, injured and control datasets were processed separately. For each dataset, raw gene UMI counts per nuclei were recorded in a matrix format before being used to create a *Seurat* object, followed by log normalization using *NormalizeData* function with setting parameters *normalization.method = "LogNormalize", scale.factor = 10,000*. To improve the signal-to-noise ratio, and produce robust and reliable results for downstream analyses, we identified highly variable genes (HVGs) using *Seurat FindVariableFeatures* function by choosing parameters *selection.method = "vst", nfeatures = 3000*. To achieve this goal, we fit the relationship between log-transformed variance and log-transformed mean of gene UMI counts using local polynomial regression, and then chose the 3000 top-ranked genes as HVGs, which were scaled subsequently using *ScaleData* to feed the request from gene dimension reduction and cell clustering. For gene dimension reduction, the first step used principle component analysis (PCA) via a linear dimension reduction approach, which was implemented by using the *Seurat RunPCA* function and consequently choosing the top 50 principal components (PCs) to represent cell variation. The second step used uniform manifold approximation and projection (UMAP)[90] to complete further dimension reduction of the top 30 PCs generated in the first step. Finally, we used the *Seurat FindClusters* function with parameters (i.e., *dims = 1:30, resolution = 1*) to analyze nuclei clustering. This method was used to identify major nuclei clustering that would ultimately be associated with the main cell type. Subsequent clustering to look for subclusters within cell types was not performed for this study.

**Assignment of cell type**. In order to identify and isolate astrocytes, nuclei clusters were matched to specific cell types by calculating gene specificity scores for each gene in each cluster using the R script retrieved from Efroni, Ip[91] or using *FindMarkers* with the *test.use = "roc"* parameter[89]. Genes with the highest specificity scores or roc scores suggested enrichment and were consequently considered as cluster-specific markers, which would be compared with the literature of reported cell type markers to assign a cell type to each cluster. Cell-type markers were as follows: astrocyte (*SLC1A2, SLC1A3, ALDH1L1, GFAP, AQP4, NDRG2*), endothelial cell (*FLT1, IGFBP7, SLC2A1*), excitatory neuron (*SYT1, RBFOX3, GRIN2A, SATB2, CUX1, CUX2*), inhibitory neuron (*GAD1, GAD2, LHX6, SST, VIP*),

microglia (*P2RY12, CD74, DOCK8, GPR34, C1QB*), oligodendrocyte (*PLP1, MBP, UGT8, ST18, MOBP*), and oligodendrocyte precursor cell (*PDGFRA, PCDH15, MEGF11*). Nuclei clusters enriched with a particular set of markers were considered to be of the corresponding cell type.

**Removal of unwanted batch effects**. In order to reduce unwanted batch effects that originated from external systematic or technical issues, we utilized Fast Mutual Nearest Neighbors (*fastMNN*)[92] correction. Here, we considered the three different injured marmosets as the potential source of external batches, since the sample dissection and library preparation were separately executed; there was also the possibility that individual differences could be a contributor. Instead of using fastMNN directly, we used the *RunFastMNN* function, which is efficiently integrated in the *SeuratWrappers*[93] package, to unite this analysis to the whole pipeline. Additionally, the execution of the *RunFastMNN* function was based on some HVGs, which were identified separately but shared by the three batches, using *Seurat FindVariableFeatures* function with parameters *selection.method = "vst", nfeatures = 3000*.

**Astrocyte-specific analysis**. Given the goal was to compare astrocytes between injured and control cohorts, the astrocyte nuclei cluster was isolated from each dataset using the *Base Subset* function[89]. The *Base Merge* function and *Seurat FindIntegrationAnchors* and *IntegrateData* functions with *anchor.features = 3000, dims = 1:30 and doms = 1:30* parameters, respectively, were then used to combine the injured and control astrocyte clusters into one Seurat object. The *Seurat FindVariableFeatures* function was used with *mean.function = ExpMean, dispersion.function LogVMR, nfeatures = 3000, selection.method = "vst"* parameters prior to integration. The object was re-scaled via the *Seurat ScaleData* function and ran through PCA using the *Seurat RunPCA* function with default and *npcs = 30* parameters, respectively, and the *Seurat JackStraw* and *ScoreJackStraw* functions with *num.replicate = 100, dims = 30 and dims = 1:30* parameters, respectively, were used after merging and integration. Importantly, the top 30 PCs were used in the *Seurat RunPCA* function and significant PCs were used for subsequent dimensional reduction with *Seurat RunUMAP* using parameters *reduction = "pca", dims = 1:X*, where X was the number of PCs with a *p*-value < 0.01 from the *Seurat JackStrawPlot*, for visualization.

Differential expression analyses of cells between injured and control cohorts was conducted using the *Seurat FindMarkers* function with *logfc.threshold = -Inf, min.pct = -Inf* parameters was adopted to identify differentially expressed genes (Supplementary Data 1). For detecting top differentially expressed gene candidates, we defined statistically significant as genes that had greater than 0.25 log-transformed fold change between groups in either direction and exhibited adjusted *p*-values (False Discovery Rate) less than 0.01; further, only genes that were expressed by at least 10% of nuclei in either population were considered. The generated data was then visualized using the *ggplot* function[94].

The top 30 upregulated differentially expressed genes (DEGs) were used for gene ontology (GO) enrichment analysis via PANTHER[95] (Supplementary Data 1). More in depth analysis of the gene of interest, *RTN4A* (NogoA), was performed using HumanBase Functional Module Detection[96]. Briefly, the *Seurat SubsetData* function was used to isolate *RTN4A*- nuclei *subset.name = "RTN4A", low.threshold = 1* and *RTN4A* + nuclei *subset.name = "RTN4A", high.threshold = 1*. DEGs for these subsets were identified using the *FindMarkers* function with *logfc.threshold = 0.25, min.pct = 0.1, only.pos = T*, parameters. The top 100 upregulated DEGs from each subset were used for the HumanBase Functional Module Detection (Supplementary Data 1).

**Fresh and fixed marmoset tissue collection for protein analyses**. At the end of the designated post-stroke period: 1 DPI, 7 DPI, 14 DPI, or 21 DPI; marmosets were administered an overdose of pentobarbitone (100 mg/kg; IM). For fresh tissue, brains were immediately harvested following apnea. The occipital poles were dissected at the level of the diencephalon and bisected coronally, with caudal portions encompassing V1 and the ischemic zone and core, before snap freezing them in liquid nitrogen.

For fixed tissue, animals were similarly euthanized and transcardially perfused with 0.1 M heparinized saline solution (0.9% sodium chloride at 37 °C containing 0.1 M heparin), followed by 4% paraformaldehyde. Brains were collected, post-fixed for 24 h in 4% paraformaldehyde, sucrose protected, and cryosectioned[2,97].

**Three days post-MCAO mouse tissue**. C57Bl6/J mouse cortical tissue (*n* = 3) was gifted from Professor Christopher G. Sobey. 8–12-week-old mice were subjected to transient MCAO for 1 h and brains were collected 72 h post-ischemia. Unperfused fresh brains were snap frozen over liquid nitrogen and sectioned onto slides at 10 μm thick and 420 μm apart before storing at −80 C. Sections were post-fixed in 4% PFA before immunostaining as described in immunolabeling protocol below. Experiments were approved by the Monash University Animal Ethics Committee.

**Human tissue**. Snap-frozen and paraffin-fixed human brain samples (*n* = 1; Age: 74 yrs; Gender: Female; COD: Cirrhosis; Post-mortem processing: <24 h) were obtained from the Newcastle Brain Tissue Resource (UK) with informed consent.

Informed consent was provided by donors or their nominated representatives under the understanding that donated brain and/or spinal cord tissue will be used to make a diagnosis and for ethically approved studies under the custodianship of the Newcastle Brain Tissue Resource (UK). Consent can be withdrawn by the donors or their nominated representative at any time before the donation without reason. Consent can be withdrawn by nominated representatives at any time after the donation without reason, which must result in the immediate destruction of the donor samples in a lawful and respectful manner. Procurement and use of human brain tissue were approved by the Monash University Human Research Ethics Committee in compliance with section 5.1.22 of the National Statement on Ethical Conduct in Human Research (Ethics approval number: CF14/2120-2014001121).

### Cell culture

*Astrocytes.* Astrocytes were obtained from mouse, marmoset, and human. For mouse astrocytes, magnetic isolation of ASCA-2+ astrocytes was performed. ASCA-2+ astrocytes were purified by immunopanning from seven P5 mouse pup cortices. Briefly, cortices were dissociated mechanically (using a scalpel blade) and enzymatically (using Papain: Worthington, LS003126) with various triturations and Nitex (Sefar, 03-20/14) filtration steps to obtain a single-cell suspension. The cell suspension was subsequently allowed to recover in DPBS (Life Technologies 14287080) containing 0.2 mg/ml BSA and 0.004 mg/ml DNaseI at 37 °C for 30–45 min in a 10% $CO_2$ incubator. The cell suspension underwent debris (Miltenyi, 130-109-398) and red blood cell (Miltenyi Cat. 130-094-183) removal before incubation to positively select for astrocytes using anti-ASCA-2 MicroBead Kit with FcR Blocking Reagent (Miltenyi, 130-097-679) and an MS column/ MACS separator. Isolated astrocytes were cultured at 37 °C in a 10% $CO_2$ tissue culture incubator in complete Astrocyte Growth Medium (AGM) as described in Lidde-low, Guttenplan[40]. The culture was maintained by replacing 50% of the medium every 7 days with complete AGM containing fresh 5 ng/mL HB-EGF. Experiments were approved by the Monash University Animal Ethics Committee.

For marmoset astrocytes, postnatal day 14 marmoset V1 was processed[98]. Astrocytes were generated by passaging neurospheres three times to maximize astrocyte yield, followed by 1-week in NeuroCult NSA with differentiation supplement (StemCell Technologies) and further expanded in Neurocult NSA with proliferation supplement (StemCell Technologies; 10 ng/ml rhFGF-2, 20 ng/ml rhEGF) and incubated at 37 °C, 5% $CO_2$.

For human astrocytes, a primary astrocyte cell line was obtained from Lonza. Human astrocytes were maintained in AGM and treated in DMEM serum-free conditions with IL-6 and IL-6 receptor for 24 h to induce a more reactive phenotype.

*Macrophages.* Human macrophages were derived from the THP-1 monocytic cell line (ATCC® TIB202™) as per manufacture instructions. Human macrophages were cultured in RPMI 1640 medium (ATCC® 302001™), supplemented with 0.05 mM 2mercaptoethanol and 10% fetal bovine serum (FBS) in antibiotic-free conditions. THP-1 cells were then differentiated into macrophages with 0.1 µM 1α,25-dihy-droxyvitamin D3 (vitamin D3; Sigma-Aldrich) or 200 nM phorbol 12-myristate 13-acetate (PMA; Sigma-Aldrich) over 72 h[38]. Cells were then treated with 200 ng/mL of recombinant rat NogoA (1026-1090aa) and Fc Chimera protein (Nogo-66; R&D Systems), or with human NogoA (566-748aa) and Fc chimera protein (Nogo-Δ20; R&D Systems), for 10 m, 30 m, 1 h, and 2 h.

### Immunohistochemistry and immunofluorescence

For immunohistochemistry (IHC), free-floating sections representing V1 were treated with 0.3% hydrogen peroxide and 50% methanol in 0.01 M PBS for 30 m to inactivate endogenous peroxidases before pre-blocking. 15% normal horse serum (NHS; Gibco, ThermoFisher, USA) in 0.01 M PBS, 0.3% TritonX-100 (PBS-TX; Sigma-Aldrich) was used to pre-block tissue for IHC and immunofluorescence (IF) before incubation with primary antibodies (Supplementary Table 1) overnight at 4 °C. For secondary labeling sections were incubated with either biotinylated secondary antibodies or Alexa Fluor donkey anti-host secondary antibodies for 1 h at room temperature. Following secondary incubation, sections for IHC were treated with streptavidin-horseradish peroxidase conjugate (GE Healthcare, Amersham, UK; 1:200) prior to visualization using metal enhanced chromagen, 3,3'-diaminobenzidine (DAB: Sigma-Aldrich). For IF staining, sections were treated with Hoechst 333258 nuclei stain, mounted on Superfrost slides and treated with 0.05% Sudan Black in 70% ethanol for 10 m before coverslipping using fluoromount-G.

For human tissue, sections were dewaxed in xylene before serial rehydration in ethanol. Antigen retrieval was performed using 0.05% citraconic anhydride at 90 °C for 2 h. Sections were subsequently blocked in 5% normal goat serum (NGS; Gibco, ThermoFisher, USA), 1% bovine serum albumin (BSA; Sigma-Aldrich) and 0.1% fish skin gelatin (Sigma-Aldrich) in 0.01 M PBS, 2% TritonX-100 (PBS-TX; Sigma-Aldrich) for 1 h before incubation with primary antibodies: rabbit anti-NogoA (1:200; Supplementary Table 1) and mouse anti-GFAP (1:500; Sigma-Aldrich), for 72 h at 4 °C. Secondary labeling was performed using goat anti-rabbit (Alexa Fluor 350; ThermoFisher) and donkey anti-mouse (Alexa Fluor 647; ThermoFisher) overnight at 4 °C before coverslipping using fluoromount-G.

For cell culture staining, cells were fixed on poly-ornithine (1 mg/mL) and laminin-coated (1 mg/mL) glass coverslips or chamber slides in 4% PFA before blocking in 10% NHS in 0.01 M PBS, 0.3% PBS-TX. Cells were incubated with

primary antibodies (Supplementary Table 1) for 2 h at room temperature before secondary labeling using Alexa Fluor donkey anti-host secondary antibodies for 1 h at room temperature. Cells were treated with Hoechst 333258 nuclei stain and coverslipped using fluoromount-G.

Stained tissue/cells were analyzed using either the Axioimager Z1 Upright Microscope and Axiovision Software (Carl Zeiss), or the Sp5 Inverted Confocal Microscope, and LAS Software (Leica Microsystems). ImageJ/ Fiji, Adobe Photoshop and Illustrator CC 2017 (Adobe) were used for image post-processing and figure design.

For Fig. 4f the representative pseudo colored LILRB2 and NogoA photomicrographs were obtained by DAB immunohistochemistry and overlaid using Adobe Photoshop in order to show spatial distribution of labeling. Please note that these photomicrographs are from different (adjacent) sections and have been overlaid to create a representative image as double labeling with LILRB2 and NogoA was not possible at the time due to available antibodies.

### Protein extraction, SDS-PAGE, and immunoblot

Snap-frozen V1 tissues were homogenized in either TRIzol LS Reagent (ThermoFisher), or NP40 (Thermo-Fisher) lysis buffer, and protein extracted according to manufacturer's instructions. Subsequent lysates were supplemented with 10% Protease Inhibitor Cocktail (PrIC; Sigma-Aldrich), 1% Phosphatase Inhibitor Cocktail 2 (PhIC; Sigma-Aldrich), 1 mM PMSF Serine Protease Inhibitor (Sigma-Aldrich). Protein concentration was determined using Bradford Reagent (Sigma-Aldrich). At experimental endpoints, cells were lysed using RIPA buffer (ThermoFisher) and addition of previously mentioned inhibitor cocktails. Marmoset and human frontal cortical tissues were also lysed in this way for use as positive controls. Equal concentrations of samples were added to 4X Loading Buffer (240 nM TRIS, 8% SDS, 40% glycerol, 20% 2-mercaptoethanol, 0.05% bromophenol blue), heated at 95 °C for 10 min, and electrophoretically separated on a 4–12% Bis-Tris gel (ThermoFisher). Following gel electrophoresis, proteins were blotted onto PVDF membranes (ThermoFisher), pre-blocked in Odyssey Blocking Buffer (PBS; Li-Cor) before incubation with primary antibodies (Supplementary Table 1) overnight at 4 °C. Following washes, membranes were incubated with IRDYE secondary antibodies (Li-Cor) and visualized using the Odyssey CLx Scanner (Li-Cor).

### Densitometry

Densitometric analysis was performed on scanned immunoblots using Image Studio Lite (Li-Cor: version 5.2.5), and values representing protein over loading control were calculated. Data distribution was evaluated for Gaussian distribution using the Shapiro–Wilk normality test. Ordinary one way-ANOVA with Dunnett's multiple comparisons post-hoc test were used for statistical analysis of densitometry data—Prism 7 (GraphPad). ImageJ/ Fiji (Rasband) and Adobe Illustrator CC 2017 (Adobe) were used for post-processing of immunoblot images and figure design.

### Antibody characterization

NogoA and LILRB2 antibodies (Supplementary Table 1) were tested in rat, marmoset and human cortical tissue to ensure uni-formity of detectable bands across species (Supplementary Fig. 6a). NogoA anti-body specificity was validated using two antibodies recognizing separate epitopes on the NogoA ligand due to the ambiguity of some of the data surrounding NogoA expression in various CNS injury models. The NogoA antibodies were tested on fresh and fixed marmoset cortical tissue, rat cortical tissue as a positive control and marmoset liver as a negative control, with and without pre-incubation with blocking peptides to determine antibody specificity (Supplementary Fig. 6b, c). TMEM119 and LILRB2 antibodies (Supplementary Table 1) were tested in fixed marmoset spleen and brain tissue with an Iba1 co-label to ensure specificity of TMEM119 to microglia in marmoset (Supplementary Fig. 6d).

### NogoA+/GFAP+ cell quantification

Z stack images were taken at high magni-fication of randomly selected GFAP+ cells within 500 µm of the ischemic core in 3 days post-MCAO mouse tissue ($n = 3$) and in 7 days post-ischemia marmoset tissue ($n = 4$). Colocalization of NogoA and GFAP or lack thereof was determined using ImageJ/Fiji and subsequently quantified as the number of NogoA+ astrocytes over the total number of counted astrocytes (GFAP+ cells) and expressed as a percentage. A minimum of 30 cells per animal, over 3–5 sections across three mice and four marmosets, were imaged and counted. These images were also used to determine the fluorescent intensity of NogoA expression by GFAP+ astrocytes. Ten cells were randomly selected from each of the three mice and four marmosets, and the mean fluorescent intensity of NogoA was calculated over the mean fluorescent intensity of GFAP using ImageJ/ Fiji and expressed as a ratio of NogoA: GFAP. Data distribution for both cell counts and fluorescent intensity were eval-uated for Gaussian distribution using the Shapiro–Wilk normality test. Unpaired t-tests were used for statistical analysis of data using Prism 7 (GraphPad).

### Iba1/TMEM119 quantification

Iba1 and TMEM119 DAB-immunolabeled mar-moset V1 tissues across control ($n = 2$), 1 ($n = 2$), 7 ($n = 3$), 14 ($n = 2$) and 21 ($n = 3$) DPI time points were imaged and photo-merged using Adobe Photoshop. The area of Iba1 and TMEM119 immunolabeling was quantified over three sec-tions from each animal using ImageJ. Data distribution was evaluated for Gaussian distribution using the Shapiro–Wilk normality test. Ordinary two way-ANOVA

and Sidak's multiple comparisons post-hoc test were used for statistical analysis of data using Prism 7 (GraphPad). For tabulated statistics see the Source Data File.

**Marmoset and human blood-derived macrophages for flow cytometry**
*Isolation of marmoset blood mononuclear cells.* Marmoset blood was collected by cardiac puncture using syringes pre-coated with sodium citrate solution (3.8% w/v) and expelled into citrate coated collection tubes (~10% v/v of citrate solution: blood volume). Mononuclear cells were collected by density centrifugation using lymphocyte separation media (1.077 g/mL, Lonza)[99]. Red blood cells were removed by lysis using $NH_4Cl$ lysis buffer, the MNC washed twice with PBS (0.5% BSA), filtered through a 40 μm strainer and counted.

*Isolation of human blood mononuclear cells.* Human blood was obtained from healthy donors (Australian Red Cross Blood Service, Melbourne) in citrate coated bags. Low density mononuclear cells (MNC) were isolated by discontinuous density centrifugation using Ficoll-Hypaque (1.077 g/mL, Pharmacia Biotech). Red blood cells were removed by lysis using $NH_4Cl$ lysis buffer, the MNC washed twice with PBS (0.5% BSA), filtered through a 40 μm strainer and counted using a CELL-DYN Emerald hematology analyzer (Abbott). All experiments were undertaken following informed consent from donors and approval from the Monash human ethics committee.

**Autologous marmoset and human serum.** To harvest autologous serum, freshly harvested non-fractionated marmoset or human blood was centrifuged at $400 \times g$ and plasma collected. The remaining red/white blood cell layer was processed for MNC the next day as described above. The collected plasma was calcified with 20% (w/v) $CaCl_2$ solution in $H_2O$ at a ratio of 1:100 and 1:20 v/v $CaCl_2$ solution: plasma for human and marmoset plasma, respectively. The mixture was left undisturbed overnight at 4 °C, centrifuged at $2000 \times g$ to remove fibrinogen clots and the serum collected and stored in aliquots at −20 °C.

**In vitro culture of monocyte-derived macrophages.** Marmoset or human MNC were resuspended in DMEM (0.5% BSA) at a density of $3 \times 10^6$ cells/mL and seeded into 12 or 24-well tissue culture plates. The cells were incubated in a $CO_2$ incubator at 37 °C for 1 h prior to non-adherent cells being removed by aspiration, leaving behind adherent monocytes. Fresh DMEM (supplemented with 10% autologous serum and Gibco GlutaMAX) was added and the adherent cells cultured for 5 days (human) or 7 days (marmoset) with autologous serum/media being replaced on day 3 and 5. The plates were gently washed with warm PBS, the adherent macrophages lifted with trypLE Express (Gibco) cell dissociation enzyme and washed with PBS (0.5% BSA) for subsequent analysis.

**Flow cytometry.** For analysis of LILRB2 expression on marmoset and human monocyte-derived macrophages were first blocked with human TruStain FcX Fc blocking solution (Biolegend), then labeled with and without anti-human LILRB2 (5 μg/mL; R&D Systems). The cells were washed with PBS (0.5% BSA) and then labeled with goat anti-mouse IgG2a-AF647 (2 μg/mL) secondary antibody. Finally, the cells were washed and labeled with a cocktail containing anti-human CD11b-PE (clone D12; BD Biosciences) and anti-human CD14-PECy7 (clone M5E2; BD Biosciences), washed, resuspended in PBS (0.5% BSA plus propidium iodide) and analyzed on an LSR II (BD Biosciences)[100]. Flow cytometric analysis was performed using FlowJo X software. The expression of LILRB2 was assessed on macrophages, which were identified as CD14+/CD11b+ cells that had been pre-gated on single (FSC-H vs FSC-A), nucleated (SSC-A vs FSC-A), viable (PI^neg) cells. For gating strategy, see Supplementary Fig. 7.

**Stripe (attraction/repulsion) assay.** Thirteen millimetre diameter glass coverslips were cleaned overnight in 2 M HCl at 60 °C followed by sonication in graded alcohols. Coverslips were then coated overnight with Poly-L-Ornithine (1 mg/mL) at 37 °C. Coverslips were dried and placed coated-side down on a silicon matrix with 50 μm wide, 50 μm apart alternating grooves (obtained from Dr. Martin Bastmeyer, Karlsruhe University, Germany). Nogo-66 (10 μg/mL), Nogo-Δ20 (10 μg/mL), or recombinant human fc (10 μg/mL) were injected into the constructs and incubated for 1.5 h at room temperature in order to create the stripes. PBS was injected to flush and wash out unbound Fc proteins and the coverslips removed from the matrix and placed face-up in a 24-well plate. Coverslips were subsequently washed in PBS followed by laminin (1 mg/mL in MEM) for 2 h at 37 °C. Human THP-1 macrophages were either untreated or treated with LILRB2 blocker antibody (10 μg/mL) for 30 m at room temp in 10% serum-rich RPMI with growth media supplements. After incubation cells were spun down and resuspended in serum-free RPMI with growth media supplements prior to loading. Coverslips were rinsed in MEM and appropriately conditioned macrophages were plated on each coverslip to create three different experimental conditions: control-fc stripes with human macrophages, Nogo-66 or Nogo-Δ20 stripes with human macrophages and Nogo-66 or Nogo-Δ20 stripes with LILRB2 blocker-treated human macrophages. The cultures were stopped with 4% PFA after at least 16 h. Stripes and macrophages were visualized using goat anti-Fc and rabbit anti-Iba1 and appropriate

secondary antibodies, respectively. NogoA-repelled Iba1+ macrophages were counted when at least 50% of the cell body was located in the space between stripes. Receptor specificity of the NogoA-dependent cell-repulsion was confirmed through LILRB2 receptor blockade using a monoclonal blocking antibody. Data distribution was evaluated for Gaussian distribution using the Shapiro–Wilk normality test. Ordinary one way-ANOVA with post-hoc Tukey's multiple comparisons tests were used for statistical analysis of cell count data using Prism 7 (GraphPad). For tabulated statistics see the Source Data File.

**Quantification and statistical analysis.** Statistical details for each experiment, including *n* numbers and the statistical test performed, and data presentation can be found in the corresponding figure legend, above in the experimental methods, or the Source Data File. Statistical analysis was performed using Prism 7 (GraphPad) software.

**Reporting summary.** Further information on research design is available in the Nature Research Reporting Summary linked to this article.

## Data availability
All sequencing data that support the findings of this study have been deposited in the National Center for Biotechnology Information Gene Expression Omnibus (NCBI GEO) and are accessible through the GEO Series accession number GSE179141. Source data are provided with this paper.

## Code availability
No custom code for analysis was utilized in this study, only custom in-house parameters.

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

## Acknowledgements

The authors wish to acknowledge the technical assistance and contributions of A.L. Chan, A. Grubman, J. Legrand, A.M. Tichy, M. de Souza, T.A. Hoang, J. Homman-Ludiye, and C.G. Sobey. This work was supported by grants from the NHMRC (APP108197) to J.A.B. at Monash University, and from the NIH (R35 NS097283) and the Falk Medical Research Trust to S.M.S. at Yale University. A NHMRC Senior Research Fellowship (APP1077677) supports J.A.B. An Australian Postgraduate Award Scholarship supports A.G.B. The Australian Regenerative Medicine Institute is supported by grants from the State Government of Victoria and the Australian Government.

## Author contributions

A.G.B., L.T., S.M.S. and J.A.B. conceived and designed the research. A.G.B. performed most of the experiments, analyzed the data and wrote the first draft of the manuscript. A.G.B., J.S., M.L. and M.S. performed the snRNAseq and analysis and wrote transcriptomic relevant sections of the manuscript. A.G.B., J.A.B. and L.T. performed the ET-1-induced stroke surgeries. B.C. performed blood work and flow cytometry. W.C.K. performed human tissue immunolabeling and assisted with surgeries. T.D.M. assisted with the ASCA2+ astrocytes isolation from mouse pups. S.M.S., N.S. and S.K.N. provided intellectual advice and support. L.T. and J.A.B. supervised the project. J.S., L.T., S.M.S. and J.A.B. were involved in manuscript editing.

## Competing interests

S.M.S. is a founder and equity holder in ReNetX Bio, Inc., which seeks clinical development of NgR1-FC (AXER-204) for spinal cord injury. The remaining authors declare no competing interests.
