## [Peer Review File · Nature Communications]

Reviewers' Comments:

Reviewer #1:

Remarks to the Author:

Studies in primates are often of crucial value on the way of a project or paradigm toward clinical translation. Here, Boghdadi et al. characterize astrocyte responses in visual cortex gray matter after a local ischemic stroke in marmosets by profiling their transcriptome 7 d. after injury. A number of plasticity and inflammation related genes and gene networks were found to be upregulated, among which the membrane protein Nogo-A (also in human strokes, but much less so in rodents!). Via the known Nogo-A receptor LILRB2/PirB Nogo-A restricts macrophage migration into the peri-infarct parenchyma as shown by knock-out experiments. All these observations have relevance for our understanding of the degenerative and repair processes at sites of CNS tissue destruction e.g. after stroke, as well as for the planned and currently on-going clinical trials with anti-Nogo-A therapeutic agents. They thus address a basic neuroscience audience as well as clinical neuroscientists.

The following questions should be considered in revising the manuscript:

1. Where precisely in the visual cortex where the astrocyte nuclei for the transcriptomic analysis sampled: All layers of the gray matter? How close to the stroke core? How restricted to the peri-infarct ischemic area/penumbra?
2. Two main populations of reactive astrocytes are described: RTN4/Nogo-A-positive and RTN4-negative. Is there any correlation of these two types of cells with their location with regard to the stroke core, the penumbra, the infiltration zone...?
3. Fig. 3 shows a comparison of mouse and marmoset astrocytes. The primate astrocytes appear much larger in size, in addition to being strongly Nogo-A-positive. Is this typical and real or a sampling artifact?
4. A description of how far the upregulation of Nogo-A (and presumably several of the other 600 stroke-regulated astrocyte genes) spreads through the cortex is lacking. Fig 4f suggests a very wide distribution, much wider than the penumbra. What about contralateral areas? Please comment.
5. The observation that also the Nogo-A fragment d20, which binds to receptors other than NgR1 or PirB/LILRB2, inhibits macrophage adhesion/migration is very interesting. The discussion of this should take in to account that very little is known on the detailed composition of the Nogo-A receptor complexes. In analogy to e.g. neurotrophin or Wnt receptors, there is a good chance that different subunits form a multisubunit heteromeric complex. Thus, antibodies against one subunit could easily sterically hinder the binding of other subunits to a different ligand site (e.g. anti-LILRB2 could block binding of d20 to S1PR2 or syndecans). Integrins have only been shown to be indirectly involved. The scheme shown in Fig.5d is misleading.
6. Macrophages have multiple functions in CNS lesions, 'good' and 'bad' ones. Effects of restricting their access to the CNS will depend on the time point after injury, the subpopulation of macrophages etc.
7. Minor point: line 466: Several antibodies against Nogo-A as well as receptor bodies and shRNA and KOs have been used in showing enhanced sprouting and regeneration after CNS lesions (not just IN1).

Reviewer #2:

Remarks to the Author:

This manuscript from Boghdadi and colleagues investigates the heterogeneity of astrocyte responses in a marmoset model of ischemic stroke, as well as highlighting some mechanistic studies around the putative function of Nogo-A in mediating this response by blocking entrance of peripheral macrophages into the brain following insult. This work represents, to my knowledge, the first such analysis of cell-type specific transcriptomic changes in the marmoset model. Indeed the hypothesis that 'immediate post-stroke blockade of NogoA action may exacerbate brain inflammation' is intriguing, and adds a new layer of complexity to ongoing therapeutic trials using anti-Nogo therapies in spinal cord injury - these new data add a great deal to the literature surrounding Nogo as a purely inhibitory molecule. The authors should also be commended for taking some time to compare key findings across marmoset, rodent, and human - particularly important given their affirmation that the marmoset model provides a more appropriate model of

the human ischemic response.

I do however have a few concerns with the data as presented in the manuscript, and would like clarifications or additional details to determine how they would alter the conclusions drawn from the study:

1. single nuclei RNASeq analysis - the assertion that astrocytes from stroke and control brains are distinctly different at the transcriptomic level is big surprise and may suggest the data was not anchored or integrated properly. Where there absolutely no physiological (ie. normal/control) astrocytes left following ischemic stroke? Even though V1 visual cortex was dissected out for downstream nuclei sequencing, it still seems unusual that a such a large region was collected that there are no baseline/normal astrocytes left. There needs to be more comprehensive descriptions of what analysis packages/pipelines were used to generate these final data - for instance, it seems likely that the different datasets (control/ischemia) have not been anchored properly (as stated above) - leading to an artifact of complete dataset separation. Canonical Correlation Analysis (or similar) should be used to anchor the experimental groups based on multiple genes that are shown not to change in individual cell types (e.g. for astrocytes perhaps *Aldh1l1*, *Sox9*, etc.). It would also be important to include additional QC data pertaining to gene, UMI, etc counts, and highlight how individual animal samples cluster across these small datasets - it is becoming apparent in other published under-powered datasets that the often individual samples are driving much of the reported heterogeneity and the authors would benefit from showing this is not the case with their own datasets. It is also unacceptable that raw data is not deposited at a public repository such as the NIH GEO.

2. additional to this, the samples sizes are quite low, however given the difficulty of obtaining NHP samples this should be considered in that context. The authors should discuss this limitation. Indeed, the astrocyte capture in both control and ischemic animals is among the lowest of all cell types sequenced (~5%), while the capture rate for non-astrocyte cells, particularly of oligodendrocytes that express a lot of *RTN4* (the gene that encodes Nogo-A), is much higher. It was a shame not to see these data also integrated into the manuscript - as they are much better powered for making some of the strong conclusions being drawn here.

3. Fig 1 - *RTLN4*+ cell gene expression - *GAP43* is also enriched in *RTN4*- astrocytes. Can the authors comment on it's inclusion here as an *RTN4*+ marker?

4. Supp Fig 3 - *IL6* treatment of acutely purified astrocytes - why was this only completed in human and also not in mouse/marmoset? Given the statement that astrocytes were purified and cultured from all three species, the lack of this response data was a shame. This would be important data to validate the primate/human-specific nature of this response.

5. *TMEM119* and microglia (line 255) - *TMEM119* levels have been reported to drop in microglia in the context of infection, injury, and disease. Can the authors comment on this as an alternate interpretation of the *TMEM119*-*IBA1*+ cells they labeled here. Given the authors have nucSeq data on a number of microglia (up to 26% of all collected nuclei in the stroke setting) they could mine these data to see if there are any *Iba1*+ cells in this population that do not express *TMEM119*.

6. the authors state in several places (e.g. lines 291, 479, among others) that *RTN4*/*Nogo-A* has an effect on 'BBB integrity' however this is not measured anywhere in this manuscript.

minor points:

1. at several instances throughout the manuscript the authors state that their sequencing results represent 'primate specific responses'. Can the authors show some meta-analysis that confirms this statement?

2. nucSeq methods/statements (e.g. line 89-90, methods) - please be more specific when describing analysis parameters - the 'Seurat algorithm' is insufficient information. Please describe the libraries used at each step, and highlight any changes made to base code

3. please label scales of Fig 2, panels a (presumably they are expression levels?)

4. Fig 3 - a low-mag image of the lesioned area would help to validate this regional-heterogeneity of RTN4+ astrocytes

5. Fig 5, panels j - the Nogo-d20 treated cells have a much larger morphology - is this representative of all treatment repeats, or just this image? The side expansion (including of the nuclei) suggest a profound response to this ligand

Reviewer #3:

Remarks to the Author:

This manuscript reports a study examining astrocyte responses to an ischemic stroke in the cerebral cortex of marmosets, a new world species of non-human primates.

The main observations are:

(a) Single nucleus RNA-sequencing (snRNAseq) is used to identify and compare 2107 nuclei in stroke tissue from n=3 animals with 594 nuclei from n=3 uninjured controls. From a technical perspective, the procedures are well described and appear rigorously conducted. The data look to be of good quality. The cluster-analyses look robust and the differentially expressed gene (DEG) data look reliable.

(b) From the snRNAseq data the authors noted RTN4A among the top 30 DEGs in stroke associated astrocyte nuclei. RTN4A is a molecule that broadly repulses migrating cells (first identified due to its effects on migration of a fibroblast cell line), which has been implicated in the repulsion of neurites. As the authors note, in the CNS RTN4A has been previously associated with oligodendrocytes and myelin and not with astrocytes. The authors chose to examine RTN4A expression in astrocytes further.

(c) RTN4A expressing astrocyte nuclei also exhibited high levels of immune regulatory DEGs.

(d) In vivo immunohistochemistry showed that marmoset astrocytes exhibited substantially and significantly higher levels of RTN4A protein compared with mouse astrocytes, in which expression was low. In vivo immunohistochemistry also confirmed RTN4A protein expression in human astrocytes near stroke tissue. The immunohistochemical images are of high quality and the results look specific and convincing.

(e) Macrophages express a RTN4A receptor LILRB2. A series of in vitro experiments including stripe-assay migration experiments show that macrophages with LILRB2 avoid stripes coated with RTN4A containing protein. These experiments also look well conducted, well controlled and are convincing.

Overall, from a technical perspective, the work appears well conducted and properly controlled and the main findings appear to be reliable. Nevertheless, I have concerns regarding some of the interpretations and claims made on the basis of the findings.

Specific comments and concerns:

1. The authors claim to have identified a "primate-specific" astrocyte response to ischemia. However, they do not present convincing evidence to support such a strong claim as the main statement in the title of the paper. First, it is incorrect to say that RTN4 is not expressed by murine astrocytes. Various online databases show that uninjured astrocytes express significant levels of RTN4 (See the attached image from the online Barres lab database). The authors here themselves show that RTN4 is expressed also by reactive mouse astrocytes. Although this expression is relatively lower than in marmosets, what that means is not clear (nor is investigated). The authors are correct that this RTN4 expression in mouse has not been studied, but that does not mean it does not exist and might not be important in injury responses. The degree to which RTN4 does or does not contribute to immune regulatory functions of murine reactive astrocytes has not been studied, but that also does not mean that similar functions might exist in mouse as in marmoset. In the end, the authors only examine potential roles for astrocyte RTN4 in marmosets and then speculate that mouse astrocytes are fundamentally different because their level of RTN4 is somewhat lower. This is not sufficient to make a strong claim for a "primate-specific" astrocyte response that is presented in the title. The authors do not present any experiments that directly examine mouse astrocytes and macrophages and show that they behave

differently from marmoset ones. At best the authors could discuss this as a possibility in their Discussion without claiming that this has been experimentally demonstrated. The main finding of the paper is that RTN4A (NOGOA) is expressed by astrocytes and limits peripheral macrophage infiltration after ischemic stroke in a non-human primate. The title should reflect this specific observation rather than try make a broad claim about species differences that have not been rigorously or convincingly shown. The title and text and discussion should focus on the specific findings that are made regarding RTN4, which are interesting, unexpected and important and merit publication in their own right. With an appropriate title, I think the paper would be appropriate for this journal.

2. A minor point: The Discussion starts off with the statement that they provide the first dataset on single cell transcriptome changes in NHP astrocytes after brain injury. Quite frankly, claims of priority of demonstration like this are at best tedious. Other high profile journals like expressly PNAS forbid them. Such claims are best omitted.

REVIEWER COMMENTS (response in red)

Reviewer #1 (Remarks to the Author):

Studies in primates are often of crucial value on the way of a project or paradigm toward clinical translation. Here, Boghdadi et al. characterize astrocyte responses in visual cortex gray matter after a local ischemic stroke in marmosets by profiling their transcriptome 7 d. after injury. A number of plasticity and inflammation related genes and gene networks were found to be upregulated, among which the membrane protein Nogo-A (also in human strokes, but much less so in rodents!). Via the known Nogo-A receptor LILRB2/PirB Nogo-A restricts macrophage migration into the peri-infarct parenchyma as shown by knock-out experiments. All these observations have relevance for our understanding of the degenerative and repair processes at sites of CNS tissue destruction e.g. after stroke, as well as for the planned and currently on-going clinical trials with anti-Nogo-A therapeutic agents. They thus address a basic neuroscience audience as well as clinical neuroscientists.

1. Where precisely in the visual cortex where the astrocyte nuclei for the transcriptomic analysis sampled: All layers of the gray matter? How close to the stroke core? How restricted to the peri-infarct ischemic area/penumbra?

Our apologies that this was not clear in the initial submission. Please refer to Figure 1a (line 136), which has been modified to clarify that the grey matter, encompassing all cortical layers of V1 operculum, including the stroke core and penumbra, was sampled. Further we have added more detail to the Methods (lines 571-583) and Results (lines 75-76).

2. Two main populations of reactive astrocytes are described: RTN4/Nogo-A-positive and RTN4-negative. Is there any correlation of these two types of cells with their location with regard to the stroke core, the penumbra, the infiltration zone...?

NogoA expression tapers off at the V1/ V2 border with decreasing expression of NogoA on GFAP+ astrocytes (lines 204-206). Please refer to new figure, Supplementary Figure 2 (line 1396).

3. Fig. 3 shows a comparison of mouse and marmoset astrocytes. The primate astrocytes appear much larger in size, in addition to being strongly Nogo-A-positive. Is this typical and real or a sampling artifact?

Yes, this is normal. Primate astrocytes are much larger in size when compared to mouse astrocytes. See data below from a recent publication of our lab. This observation was added to Results (lines 230-232).

Jihane Homman-Ludiye, James A Bourne, The Marmoset: The Next Frontier in Understanding the Development of the Human Brain, *ILAR Journal*, 2021; ilaa028, <https://doi.org/10.1093/ilar/ilaa028>

4. A description of how far the upregulation of Nogo-A (and presumably several of the other 600 stroke-regulated astrocyte genes) spreads through the cortex is lacking. Fig 4f suggests a very wide distribution, much wider than the penumbra. What about contralateral areas? Please comment.

Please refer to new figure, Supplementary Figure 2 (line 1396), for macroscopic photomicrographs which better demonstrate the extent of the NogoA+ astrocyte distribution within the ischemic zone and beyond. Contralateral areas were not analyzed as this was outside the scope of this paper.

5. The observation that also the Nogo-A fragment D20, which binds to receptors other than NgR1 or PirB/LILRB2, inhibits macrophage adhesion/migration is very interesting. The discussion of this should take into account that very little is known on the detailed composition of the Nogo-A receptor complexes. In analogy to e.g. neurotrophin or Wnt receptors, there is a good chance that different subunits form a multisubunit heteromeric complex. Thus, antibodies against one subunit could easily sterically hinder the binding of other subunits to a different ligand site (e.g. anti-LILRB2 could block binding of d20 to S1PR2 or syndecans). Integrins have only been shown to be indirectly involved. The scheme shown in Fig.5d is misleading.

Figure 5d (line 392) has been modified to include both NgR1 and S1PR2 but are grayed out given we have demonstrated their absence on human macrophages. The schematic in Figure 5d (line 392) is the simplest model and it is possible that other NogoA receptors could be involved and be directly or indirectly inhibited by antibodies as a result of cross-subunit steric effects.

This point has been added to the Results (lines 378-384).

6. Macrophages have multiple functions in CNS lesions, 'good' and 'bad' ones. Effects of restricting their access to the CNS will depend on the time point after injury, the subpopulation of macrophages etc.

We completely agree with this comment. As highlighted in the Discussion (lines 509-510), macrophages are crucial for the clearance of detrimental cell and myelin debris after CNS injury. Astrocyte corralling of peripheral macrophages is an essential function following ischemia, evidenced by experimental data demonstrating worse functional outcomes when abolishing this function (lines 511-513). We hypothesize that astrocyte corralling at 7 DPI is a positive interaction, keeping the macrophages where they need to be to clean up and stabilize the injury site. In the acute stages post-ischemic stroke, peripheral macrophages are a mix of pro- and anti-inflammatory phenotypes, with anti-inflammatory phenotypes dominating (1-3). However, over time, there is a downregulation of anti-inflammatory genes whereas pro-inflammatory genes persist (1-3). Akin to others (4), we do not think limiting peripheral macrophage infiltration is a viable strategy following stroke. Ideally, we would want to find a way to limit the persistence of a pro-inflammatory phenotype more chronically.

1. Jian Z, *et al.* The Involvement and Therapy Target of Immune Cells After Ischemic Stroke. *Frontiers in Immunology* **10**, (2019).
2. Gliem M, Schwaninger M, Jander S. Protective features of peripheral monocytes/macrophages in stroke. *Biochimica et Biophysica Acta (BBA) - Molecular Basis of Disease* **1862**, 329-338 (2016).
3. Planas AM. Role of Immune Cells Migrating to the Ischemic Brain. *Stroke* **49**, 2261-2267 (2018).
4. Schmidt A, *et al.* Targeting Different Monocyte/Macrophage Subsets Has No Impact on Outcome in Experimental Stroke. *Stroke* **48**, 1061-1069 (2017).

7. Minor point: line 466: Several antibodies against Nogo-A as well as receptor bodies and shRNA and KOs have been used in showing enhanced sprouting and regeneration after CNS lesions (not just IN1).

This point has been added to the Discussion (lines 490-492) and a review we published which included a discussion on the various NogoA-related therapeutics has been cited (Boghdadi *et al.* 2018).

Reviewer #2 (Remarks to the Author):

This manuscript from Boghdadi and colleagues investigates the heterogeneity of astrocyte responses in a marmoset model of ischemic stroke, as well as highlighting some mechanistic studies around the putative function of Nogo-A in mediating this response by blocking entrance of peripheral macrophages into the brain following insult. This work represents, to my knowledge, the first such analysis of cell-type specific transcriptomic changes in the marmoset model. Indeed the hypothesis that 'immediate post-stroke blockade of NogoA action may exacerbate brain inflammation' is intriguing, and adds a new layer of complexity to ongoing therapeutic trials using anti-Nogo therapies in spinal cord injury - these new data add a great deal to the literature surrounding Nogo as a purely inhibitory molecule. The authors should also be commended for taking some time to compare key findings across marmoset, rodent, and human - particularly important given their affirmation that the marmoset model provides a more appropriate model of the human ischemic response.

I do however have a few concerns with the data as presented in the manuscript, and would like clarifications or additional details to determine how they would alter the conclusions drawn from the study:

1. single nuclei RNASeq analysis - the assertion that astrocytes from stroke and control brains are distinctly different at the transcriptomic level is big surprise and may suggest the data was not anchored or integrated properly. Where there absolutely no physiological (ie. normal/control) astrocytes left following ischemic stroke? Even though V1 visual cortex was dissected out for downstream nuclei sequencing, it still seems unusual that a such a large region was collected that there are no baseline/normal astrocytes left. There needs to be more comprehensive descriptions of what analysis packages/pipelines were used to generate these final data - for instance, it seems likely that the different datasets (control/ischemia) have not been anchored properly (as stated above) - leading to an artifact of complete dataset separation. Canonical Correlation Analysis (or similar) should be used to anchor the experimental groups based on multiple genes that are shown not to change in individual cell types (e.g. for astrocytes perhaps Aldh111, Sox9, etc.). It would also be important to include additional QC data pertaining to gene, UMI, etc counts, and highlight how individual animal samples cluster across these small datasets - it is becoming apparent in other published under-powered datasets that the often individual samples are driving much of the reported heterogeneity and the authors would benefit from showing this is not the case with their own datasets. It is also unacceptable that raw data is not deposited at a public repository such as the NIH GEO.

The transcriptomic data has been re-integrated using an additional 1000 anchor points, which revealed greater overlap. This does not change any of the downstream data analysis, but only the UMAP & feature plots in Figure 1d (line 136) and 2a (line 174). More detailed code has been added to the Methods (lines 571-583). QC data pertaining to detected genes, UMI and counts between individual animals from injured dataset has been added to Supplementary Figure 1d-f (line 1374). "Removal of unwanted batch effects" was added to Supplementary Methods (lines 1225-1235).

The data has been added to a public repository (NCBI GEO) and following statement added to the "Data and materials availability" section of the manuscript "All sequencing data that support the findings of this study have been deposited in the National Center for Biotechnology Information Gene Expression Omnibus (NCBI GEO) and are accessible through the GEO

Series accession number **GSE179141**. All other relevant data are available from the corresponding author on request.” (Lines 1081-1085).

To review GEO accession GSE179141:

Go to <https://www.ncbi.nlm.nih.gov/geo/query/acc.cgi?acc=GSE179141>

Enter token mpsrqmgapxehfmn into the box

2. additional to this, the samples sizes are quite low, however given the difficulty of obtaining NHP samples this should be considered in that context. The authors should discuss this limitation. Indeed, the astrocyte capture in both control and ischemic animals is among the lowest of all cell types sequenced (~5%), while the capture rate for non-astrocyte cells, particularly of oligodendrocytes that express a lot of RTN4 (the gene that encodes Nogo-A), is much higher. It was a shame not to see these data also integrated into the manuscript - as they are much better powered for making some of the strong conclusions being drawn here.

While our sample size is low, and this is an important consideration with nonhuman primate tissue, our QC plots demonstrate we have collected enough nuclei from each individual marmoset, and we do not see any obvious differences among them.

We completely agree that looking at oligodendrocytes, the cell type previously associated with NogoA, would indeed be interesting. However, the focus of this particular paper is astrocytes and the unique finding that NogoA is upregulated on them following ischemic stroke. Rest assured that future research by the groups involved in this current study is underway for non-astrocyte cell types.

3. Fig 1 - RTN4+ cell gene expression - GAP43 is also enriched in RTN4- astrocytes. Can the authors comment on it's inclusion here as an RTN4+ marker?

GAP43, KLF6, CD44 were all simply chosen as markers for in tissue validation of the transcriptomic data. We are not claiming that GAP43 is a marker of RTN4+ astrocytes. GAP43 was differentially expressed within the top 100 genes for both RTN4- and RTN4+ astrocyte after injury compared to control. These genes were also chosen partially due to the HumanBase functional analysis performed earlier. GAP43 being expressed on both RTN4+ and RTN- populations gave us a better spread for the cell counts between control and injury, which is why we chose to quantify GAP43 specifically. This was performed to demonstrate the reliability of our transcriptomic data to our in-tissue data. For example, the transcriptomic data tells us that 51% RTN4+ astrocytes express GAP43 in the injured cohort compared to 6% in the control cohort. Our in-tissue validation counts revealed very similar numbers: 44% in injured and 7% in control.

4. Supp Fig 3 - IL6 treatment of acutely purified astrocytes - why was this only completed in human and also not in mouse/marmoset? Given the statement that astrocytes were purified and cultured from all three species, the lack of this response data was a shame. This would be important data to validate the primate/human-specific nature of this response.

Thanks for pointing this out. IL6 treatment has been removed (Supplementary Figure 4, line 1429) because we are not claiming a difference between rodent and primate, just that NogoA is expressed on astrocytes from all 3 species in culture, in the absence of myelin. This experiment was simply to demonstrate the expression of NogoA in the 3 species. The more important

experiment was to show that LILRB2+ human macrophages were repelled by NogoA. The fact that mouse and marmoset data was presented here was simply to show that NogoA was expressed by them. Additionally, as suggested by the reviewers of our manuscript, all references to primate specificity have been removed from the manuscript, so the treatment of human astrocytes with IL6 is no longer relevant.

5. TMEM119 and microglia (line 255) - TMEM119 levels have been reported to drop in microglia in the context of infection, injury, and disease. Can the authors comment on this as an alternate interpretation of the TMEM119-IBA1+ cells they labeled here. Given the authors have nucSeq data on a number of microglia (up to 26% of all collected nuclei in the stroke setting) they could mine these data to see if there are any Iba1+ cells in this population that do not express TMEM119.

Although some studies have reported a drop in TMEM119 levels in the context of infection, injury and disease, studies also support using TMEM119 as a marker of reactive microglia in human traumatic brain injury (1), in human multiple sclerosis grey matter lesions (2), and in injured mouse spinal cord (3).

Unfortunately, TMEM119 is not annotated in our marmoset genome alignment. Additionally, within our defined microglia (macrophage) cluster there also exists a large proportion of peripheral macrophages. Therefore, these 2 factors combined means that we were unable to look for Iba1+/ TMEM119- nuclei in our transcriptomic dataset and answer this question. Therefore, we cannot exclude that there are TMEM119- microglia at the infarct site.

A subset of this point has been added to Results (lines 262-264).

1. Bohnert S, *et al.* TMEM119 as a specific marker of microglia reaction in traumatic brain injury in postmortem examination. *International Journal of Legal Medicine* **134**, 2167-2176 (2020).
2. Van Wageningen TA, Vlaar E, Kooij G, Jongenelen CAM, Geurts JJG, Van Dam A-M. Regulation of microglial TMEM119 and P2RY12 immunoreactivity in multiple sclerosis white and grey matter lesions is dependent on their inflammatory environment. *Acta Neuropathologica Communications* **7**, (2019).
3. Zhou X, *et al.* Microglia and macrophages promote corralling, wound compaction and recovery after spinal cord injury via Plexin-B2. *Nat Neurosci* **23**, 337-350 (2020).

6. the authors state in several places (e.g. lines 291, 479, among others) that RTN4/Nogo-A has an effect on 'BBB integrity' however this is not measured anywhere in this manuscript.

As we have not undertaken any experiments to validate "BBB integrity", any reference to this has been removed from the manuscript.

minor points:

1. at several instances throughout the manuscript the authors state that their sequencing results represent 'primate specific responses'. Can the authors show some meta-analysis that confirms this statement?

Our apologies, we agree that this statement was over ambitious. The word "primate-specific" has been removed from the manuscript.

2. nucSeq methods/statements (e.g. line 89-90, methods) - please be more specific when describing analysis parameters - the 'Seurat algorithm' is insufficient information. Please describe the libraries used at each step, and highlight any changes made to base code

More detailed code has been added to the Methods (lines 571-583).

3. please label scales of Fig 2, panels a (presumably they are expression levels?)

Our apologies for this oversight. A scale has been added to Figure 2a: "Exp (norm)" (line 174).

4. Fig 3 - a low-mag image of the lesioned area would help to validate this regional-heterogeneity of RTN4+ astrocytes

NogoA expression tapers off at the V1/ V2 border with decreasing expression of NogoA on GFAP+ astrocytes (lines 204-206). Please refer to new figure, Supplementary Figure 2 (line 1396).

5. Fig 5, panels j - the Nogo-d20 treated cells have a much larger morphology - is this representative of all treatment repeats, or just this image? The side expansion (including of the nuclei) suggest a profound response to this ligand

The macrophages used in these assays are a heterogenous population and vary in size and morphology. The larger morphology observed in the Nogo-d20 representative image can also be found in Fc and blocker treatment cohorts (Figure 5j, line 392), therefore this observation does not appear to be a response to the ligand.

Reviewer #3 (Remarks to the Author):

This manuscript reports a study examining astrocyte responses to an ischemic stroke in the cerebral cortex of marmosets, a new world species of non-human primates.

The main observations are:

- (a) Single nucleus RNA-sequencing (snRNAseq) is used to identify and compare 2107 nuclei in stroke tissue from n=3 animals with 594 nuclei from n=3 uninjured controls. From a technical perspective, the procedures are well described and appear rigorously conducted. The data look to be of good quality. The cluster-analyses look robust and the differentially expressed gene (DEG) data look reliable.
- (b) From the snRNAseq data the authors noted RTN4A among the top 30 DEGs in stroke associated astrocyte nuclei. RTN4A is a molecule that broadly repulses migrating cells (first identified due to its effects on migration of a fibroblast cell line), which has been implicated in the repulsion of neurites. As the authors note, in the CNS RTN4A has been previously associated with oligodendrocytes and myelin and not with astrocytes. The authors chose to examine RTN4A expression in astrocytes further.
- (c) RTN4A expressing astrocyte nuclei also exhibited high levels of immune regulatory DEGs.
- (d) In vivo immunohistochemistry showed that marmoset astrocytes exhibited substantially and significantly higher levels of RTN4A protein compared with mouse astrocytes, in which expression was low. In vivo immunohistochemistry also confirmed RTN4A protein expression in human astrocytes near stroke tissue. The immunohistochemical images are of high quality and the results look specific and convincing.
- (e) Macrophages express a RTN4A receptor LILRB2. A series of in vitro experiments including stripe-assay migration experiments show that macrophages with LILRB2 avoid stripes coated with RTN4A containing protein. These experiments also look well conducted, well controlled and are convincing.

Overall, from a technical perspective, the work appears well

1. The authors claim to have identified a “primate-specific” astrocyte response to ischemia. However, they do not present convincing evidence to support such a strong claim as the main statement in the title of the paper. First, it is incorrect to say that RTN4 is not expressed by murine astrocytes. Various online databases show that uninjured astrocytes express significant levels of RTN4 (See the attached image from the online Barres lab database). The authors here themselves show that RTN4 is expressed also by reactive mouse astrocytes. Although this expression is relatively lower than in marmosets, what that means is not clear (nor is investigated). The authors are correct that this RTN4 expression in mouse has not been studied, but that does not mean it does not exist and might not be important in injury responses. The degree to which RTN4 does or does not contribute to immune regulatory functions of murine reactive astrocytes has not been studied, but that also does not mean that similar functions might exist in mouse as in marmoset. In the end, the authors only examine potential roles for astrocyte RTN4 in marmosets and then speculate that mouse astrocytes are fundamentally different because their level of RTN4 is somewhat lower. This is not sufficient to make a strong claim for a “primate-specific” astrocyte response that is presented in the title. The authors do not present any experiments that directly examine mouse astrocytes and macrophages and show that they behave differently from marmoset ones. At best the authors could discuss this as a possibility in their Discussion without claiming that this has been experimentally demonstrated. The main finding of the paper is that RTN4A (NOGOA) is expressed by astrocytes and limits peripheral macrophage infiltration after ischemic stroke in a non-human primate. The title should reflect this specific observation rather than try make a broad claim about species differences that have not been rigorously or convincingly shown. The title and text and discussion should focus on the specific findings that are made regarding RTN4, which are interesting, unexpected and important and merit publication in their own right. With an appropriate title, I think the paper would be

appropriate for this journal.

See highlighted text (lines 202-204): “These data are supported by previous transcriptomic analysis of reactive astrocytes where RTN4A remains unchanged in adult mice 3 days after transient middle cerebral artery occlusion (19, 20).”

See highlighted and added text (lines 459-465), which says that although NogoA is expressed on reactive astrocytes in mouse, NogoA is not upregulated at this specific time point after ischemic stroke when we have influx of blood borne macrophages.

“Primate-specific” has also been removed from the title.

2. A minor point: The Discussion starts off with the statement that they provide the first dataset on single cell transcriptome changes in NHP astrocytes after brain injury. Quite frankly, claims of priority of demonstration like this are at best tedious. Other high profile journals like expressly PNAS forbid them. Such claims are best omitted.

Our apologies, the aforementioned statement has been removed.

Reviewers' Comments:

Reviewer #1:

Remarks to the Author:

This was a very good manuscript already before the revisions, and it became really excellent now. The additional information and figures (esp. in the Suppl.) help to clarify many questions. All my points of criticism and suggestions were taken care of very well.

All in all, the study is of high relevance for our understanding of stroke pathophysiology, in particular of inflammatory and immune processes and their molecular regulation. On a basic science level, important new aspects have been added to the multiple roles of astrocytes. The focus on non-human primates of the paper, but also the comparison with rodents and human tissue are of great value.

Reviewer #2:

Remarks to the Author:

The authors have addressed all my concerns that were raised in the original round of review.

I would like to make particular mention that the expanded data analysis detail/methods is much appreciated. Also, while the downstream data output from integrated v non-integrated data are similar, I thank the authors for completing this important visualization and pre-analysis step, as it is important for the field to ensure that readers do not assume that all astrocytes lose their original transcriptomic profile - in truth it is often only a small subset.

Additional error corrections throughout make for a well-rounded and important manuscript.

I also appreciate the more thorough narrative around RTLN4+/- astrocytes and GAP43 - if there is space I think this would be a nice addition to the final manuscript.

In relation to R1 comments on marmoset astrocyte sizes - given this is a well-known difference between rodent and human astrocytes, it is nice to now see this explicitly stated for marmoset as well.

Reviewer #3:

Remarks to the Author:

I have gone through the manuscript carefully and have no additional concerns. I continue to find the results interesting.